# Implicit Modeling for Transferability Estimation of Vision Foundation Models

**Yaoyan Zheng**    **Huiqun Wang**    **Nan Zhou**    **Di Huang**[*]
[1]State Key Laboratory of Complex and Critical Software Environment,
Beihang University, Beijing, China
[2]School of Computer Science and Engineering,
Beihang University, Beijing, China
{yaoyanzheng,hqwangscse,zhounan0431,dhuang}@buaa.edu.cn

## Abstract

Transferability estimation identifies the best pre-trained models for downstream tasks without incurring the high computational cost of full fine-tuning. This capability facilitates deployment and advances the pre-training and fine-tuning paradigm. However, existing methods often struggle to accurately assess transferability for emerging pre-trained models with diverse architectures, training strategies, and task alignments. In this work, we propose Implicit Transferability Modeling (ITM), a novel framework that implicitly models each model's intrinsic transferability, coupled with a Divide-and-Conquer Variational Approximation (DVA) strategy to efficiently approximate embedding space evolution. This design enables generalization across a broader range of models and downstream tasks. Extensive experiments on a comprehensive benchmark—spanning extensive training regimes and a wider variety of model types—demonstrate that ITM consistently outperforms existing methods in terms of stability, effectiveness, and efficiency. Code is available at https://github.com/BUAAHugeGun/ITM.

## 1 Introduction

With the success of the pre-training and fine-tuning paradigm, a substantial number of pre-trained models are publicly available. However, recent studies [1] reveal that their performance varies significantly across architectures, datasets, and pre-training strategies, posing increasing challenges in identifying the most suitable models for downstream tasks.

Transferability Estimation (TE), which predicts the performance ranking of pre-trained models on downstream tasks with minimal computational cost, attracts considerable research attention as an efficient solution to model selection. Unlike the computationally prohibitive brute-force approach of individually fine-tuning each model, TE methods aim to develop efficient metrics to quantify model suitability. Early approaches estimate the compatibility between pre-trained models and downstream tasks by measuring divergences in logit spaces [2, 3] or embedding spaces [4, 5], while more recent methods improve upon static estimations by simulating the dynamic evolution of embedding spaces during fine-tuning to enhance predictive accuracy [6, 7].

However, most existing TE techniques are primarily evaluated on models with similar architectures and training paradigms (e.g., supervised pre-trained CNNs) and fail to generalize to models trained with advanced pre-training strategies or novel architectural designs. When applied to ViT [8] models developed using techniques such as Instance Discrimination (ID) [9, 10] and Masked Image Modeling (MIM) [11, 12], the distinct convergence behaviors exhibited [11, 1] further complicate unified

---

[*]Corresponding author.

39th Conference on Neural Information Processing Systems (NeurIPS 2025).

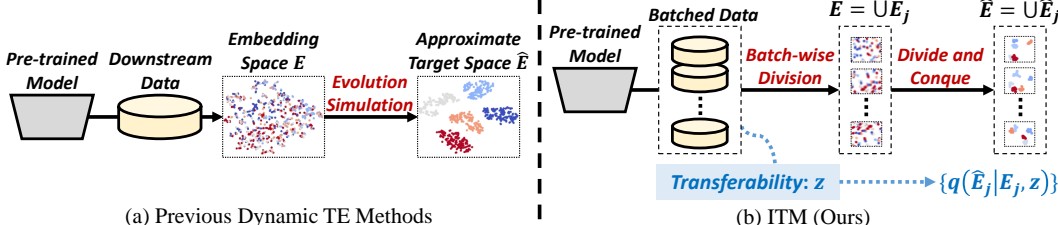

Figure 1: **Comparison between previous dynamic TE methods and the proposed ITM.** (a) Previous methods simulate the entire embedding space by hand-crafted rules and estimate transferability based on the approximated target space. (b) ITM implicitly models transferability $z$ and simulates subspace evolution using a divide-and-conquer strategy for more accurate estimation.

estimation, leading to notable declines in accuracy and unreliable model selection. Meanwhile, these methods are predominantly designed for image classification tasks and lack adaptation mechanisms for downstream task heads, thereby limiting their applicability to more complex scenarios such as semantic segmentation.

The performance of a pre-trained model on downstream tasks depends on two key aspects: (1) its intrinsic properties, such as architecture, pre-training data sources, and strategies; and (2) the characteristics of the downstream tasks, which impose varying demands on model-specific capabilities. The interplay between these factors produces unique adaptation dynamics for each model, resulting in different levels of transferability across scenarios. Although previous studies attempt to model these dynamics by directly simulating the evolution of embedding spaces, they fail to comprehensively capture both aspects, thereby limiting their generalization capabilities.

To address these limitations, we aim to incorporate an implicit modeling of a model's transferability into the simulation of embedding space evolution by using a small number of learnable parameters, enabling a more comprehensive assessment of both model properties and task adaptability. However, achieving this goal presents two major challenges: (1) implicit modeling requires knowledge of the final embedding states after fine-tuning, which is impractical for direct estimation; and (2) modeling the evolution of candidate models across downstream tasks is computationally intensive, as adaptation dynamics vary significantly across models and tasks.

In response to these challenges, we propose the Implicit Transferability Modeling (ITM) paradigm. As illustrated in Fig. 1, ITM decouples the transferability of pre-trained models and encodes them through an implicit latent representation, enabling the approximation of embedding space evolution without requiring explicit simulation. To achieve feasibility and computational efficiency, we introduce a Divide-and-Conquer Variational Approximation (DVA) that partitions the embedding space into subspaces and streamlines their evolution modeling. By integrating these components, ITM enables more precise and scalable transferability estimation across diverse architectures and pre-training strategies. We benchmark ITM on recent models following more complete training regimes across ten widely used downstream tasks and conduct comprehensive comparisons against state-of-the-art TE methods. Experimental results show that ITM consistently outperforms existing approaches, achieving stable and substantial gains in estimation accuracy and generalization ability.

Our contributions are summarized as follows:

1. We introduce the Implicit Transferability Modeling (ITM) paradigm for accurate and generalized transferability estimation across diverse architectures and pre-training strategies.

2. We propose a Divide-and-Conquer Variational Approximation (DVA) strategy to efficiently model embedding evolution while minimizing computational costs.

3. We achieve state-of-the-art performance in transferability estimation, demonstrating substantial improvements across ten downstream tasks and ten recent models trained with various pre-training methods.

## 2 Related work

### 2.1 Transferability estimation

Distinct from conventional model selection strategies based on traditional fine-tuning and its variants, such as pruning [1], task-head tuning [13], or model encoding [14], Transferability Estimation has emerged as a promising research direction, distinguished from prior approaches by its computational efficiency and simple, plug-and-play design. Based on the type of estimation method, recent TE approaches [2, 3, 4, 5, 15, 16, 6, 17, 18, 19, 20, 21, 22, 23, 24] can be categorized into two primary realms: static statistical methods and dynamic evolution-based methods.

**Static statistical methods** assess transferability by analyzing the statistical properties of the embedding spaces of pre-trained models. NCE [2] and LEEP [3] estimate logit discrepancies between model outputs and downstream task annotations using conditional probability or Bayesian metrics. $\mathcal{N}$LEEP [25] improves upon LEEP by replacing the output layer with a Gaussian Mixture Model (GMM) to enhance efficiency and calibration. LogME [4] leverages the logarithm of maximum evidence to provide stable predictions at lower computational cost. ETran [5] combines multiple metrics, introducing an energy-based measure to improve estimation accuracy, while GBC [15] employs the Bhattacharyya coefficient to evaluate class separability in the feature space. These static approaches eliminate the need for complex parameter updates, thereby achieving greater computational efficiency. However, by neglecting the embedding evolution that occurs during fine-tuning, they fail to capture full adaptation dynamics, ultimately limiting their prediction accuracy.

**Dynamic evolution-based methods** simulate aspects of the fine-tuning process through mapping functions or learning frameworks to achieve more accurate transferability estimation. PED [7] introduces a potential energy-based update model to predict the evolved state. LEAD [6] employs ordinary differential equations and downstream objectives to better capture the evolution of logits during adaptation. SA [22] perturbs the feature space through spread and attracts operations to approximate the robustness exhibited during fine-tuning. Although these methods represent significant progress, the recent proliferation of pre-trained models with diverse architectures introduces greater discrepancies in initial states and convergence behaviors. Existing dynamic approaches overlook the intrinsic properties of pre-trained models and focus solely on simulating embedding space updates, thereby limiting their ability to provide comprehensive and accurate transferability estimation.

### 2.2 Pre-trained models

The pre-training and fine-tuning paradigm leverages large-scale dataset knowledge for better performance on downstream tasks. Recent work further expands it by exploring diverse architectures and pre-training strategies, resulting in substantial progress and increasingly varied model capabilities.

**Model architecture.** Early applications demonstrate the paradigm's effectiveness on traditional convolutional neural networks (CNNs) such as ResNet [26] and DenseNet [27]. Compared to training from scratch, these CNNs learn rich prior knowledge from large-scale datasets and achieve superior downstream performance. More recently, Vision Transformers (ViTs) [8] and Swin Transformers [28] introduce transformer architectures [29] into computer vision. Unlike CNNs, ViTs better capture long-range dependencies and global context, exhibit distinct convergence behaviors, and achieve improved performance on complex downstream tasks.

**Pre-training strategy.** Early pre-training efforts adopt a fully supervised approach, with ImageNet [30] as the dominant dataset to boost model performance. However, reliance on dense human annotations limits scalability and broader applicability. More recently, contrastive learning strategies such as SimCLR [31, 32] and MoCo [33, 34] leverage large-scale unlabeled data, enabling models to acquire richer prior knowledge and achieving strong transfer learning results. In parallel, masked image modeling approaches like MAE [11] and SimMIM [12] reconstruct masked input regions, delivering state-of-the-art performance across a variety of downstream tasks.

Despite these advances, the growing diversity of pre-training strategies and model architectures leads to pre-trained models with significantly differing characteristics [1], posing challenges for existing transferability estimation methods. This highlights the need for more generalizable and robust TE approaches capable of handling a broader range of models.

## 3 Methods

### 3.1 Problem formulation

Consider a given model collection $\Phi = \{\phi_i\}_{i=1}^M$, where the models $\phi_i$ have different architectures and are pre-trained using distinct strategies. For ground-truth model ranking, every model is fine-tuned on the downstream training set $\mathcal{D}_T$ and evaluated on the test set $\mathcal{D}_{\mathcal{E}}$. The goal of transferability estimation is to predict the performance ranking of the collection of models by assigning a metric score $s_i$ to each model $\phi_i$, based on the extracted features $\mathbf{E}_i$ from $\phi_i$ and the labels $\mathbf{Y}$ of the downstream dataset, thereby quantifying each model's suitability for downstream tasks.

To evaluate the ranking correlation between the predicted scores $\mathcal{S} = \{s_i\}_{i=1}^M$ and the true performance $\mathcal{R} = \{r_i\}_{i=1}^M$, we follow prior work [6, 7, 4], adopting weighted Kendall's $\tau_w$ as the metric: $\tau_w = \frac{1}{\sum_{i<j} w_{ij}} \cdot \sum_{i<j} w_{ij} \cdot \text{sign}[(G_i - G_j)(P_i - P_j)]$, where $G_i, P_i \in [1, M]$ denote the ranks of the $i$-th element in $\mathcal{R}$ and $\mathcal{S}$, respectively, and $w_{ij} = \frac{1}{G_i + G_j}$ assigns a weight based on the importance of the model pair. The value of weighted Kendall's $\tau_w$ quantifies the consistency between the predicted and true rankings. Higher values indicate stronger alignment and better transferability estimation performance, thereby enabling more reliable model ranking in real-world application scenarios.

### 3.2 Implicit modeling of transferability

When adapting a pre-trained model $\phi_i$ to a downstream task, its final performance depends on both the intrinsic properties of $\phi_i$ and the characteristics of the downstream task $\mathcal{D}$. The fine-tuning process can be viewed as adapting a model from prior knowledge $\mathcal{K}$, learned from the pre-training data, to task-specific knowledge $\hat{\mathcal{K}}$ on downstream data [35], governed by an intrinsic attribute $\Psi(\mathcal{K}, \hat{\mathcal{K}})$ that determines transferability. Recent dynamic TE approaches primarily predict performance by modeling a mapping $\Gamma(\phi, \mathcal{D}_T) : \mathbf{E} \to \hat{\mathbf{E}}$, where $\mathbf{E}$ is the original embedding space of $\phi$ on $\mathcal{D}_T$, and $\hat{\mathbf{E}}$ is its approximate state after fine-tuning. However, methods relying on handcrafted rules or idealized assumptions only partially capture the knowledge transfer process and struggle to generalize across the growing diversity of model architectures and pre-training strategies.

To address this, we propose implicitly modeling of the transferability variable $z$ for each model-task pair, rather than directly modeling $\Gamma(\cdot, \cdot)$ as in previous work. The post-fine-tuning embedding space $\hat{\mathbf{E}}$ is represented as a posterior distribution $q(\hat{\mathbf{E}}|\mathbf{E}, z)$, transforming the complex mapping into a probabilistic estimation framework. Building on this formulation, we introduce a Divide-and-Conquer Variational Approximation (DVA) to efficiently approximate the final embedding after full fine-tuning, enabling more accurate and scalable transferability estimation across diverse pre-trained models.

### 3.3 Divide-and-conquer variational approximation

To enable effective transferability estimation with implicit modeling, we propose a Divide-and-Conquer Variational Approximation (DVA) to approximate the evolution of the embedding space.

**Batch-wise division.** Given the initial embedding space $\mathbf{E}$, a parameterized module $\psi$ aims to approximate the ideal mapping $\Gamma(\cdot, \cdot)$ by maximizing the posterior distribution $q_\psi(\hat{\mathbf{E}}|\mathbf{E}, z)$, where $z$ represents the latent variable capturing the global characteristics of the pre-trained model $\phi$ on the downstream dataset $\mathcal{D}$. Based on the principle of batch-shuffled independence assumptions [36, 37, 38], we treat the evolution of each subspace independently. Benefiting from this paradigm, we mitigate the entanglement of the global embedding space, enabling a more flexible and architecture-agnostic modeling process. This stands in contrast to prior TE methods that attempt to model the evolution of the entire embedding space, which is often highly entangled and architecture-dependent. To this end, we partition the embedding space into a collection $\mathcal{A}$ comprising $K$ subspaces, and approximate the overall mapping by independently modeling each subspace with $\psi$ as follows:

$$
\begin{aligned}
q_\psi(\hat{\mathbf{E}}|\mathbf{E}, z) &= q_\psi((\hat{\mathbf{E}}_1, ..., \hat{\mathbf{E}}_K)|(\mathbf{E}_1, ..., \mathbf{E}_K), z) \\
&= \prod_{j=1}^K q_\psi(\hat{\mathbf{E}}_j|\mathbf{E}_j, z).
\end{aligned}
\tag{1}
$$

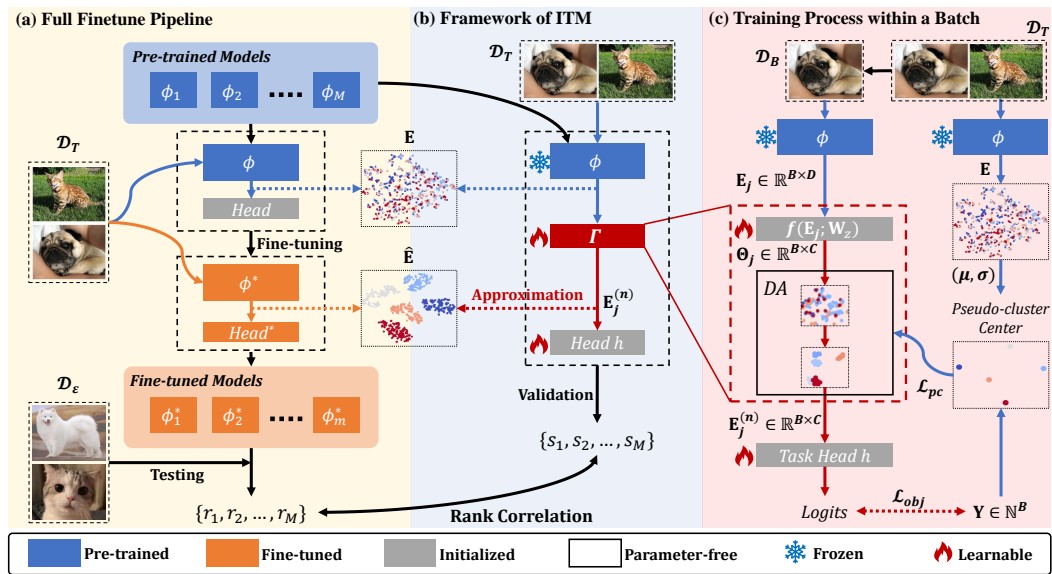

Figure 2: **Illustration of the proposed Implicit Transferability Modeling (ITM) paradigm.** (a) Ground-truth model ranking. (b) Overview of ITM, which approximates embedding space evolution via Divide-and-Conquer Variational Approximation (DVA). (c) Detailed view of DVA on a single mini-batch: the transferability $z$ is integrated into $\mathbf{E}_j$ to form the posterior condition $\boldsymbol{\Theta}_j$ via $\mathbf{W}_z$, which is jointly optimized with the downstream task head through the downstream objective $\mathcal{L}_{obj}$.

Thus, the posterior $q_\psi(\hat{\mathbf{E}}|\mathbf{E}, z)$, originally defined over the full embedding space $\mathbf{E}$, can be factorized into a series of posteriors $q_\psi(\hat{\mathbf{E}}_j|\mathbf{E}_j, z)$ corresponding to subdivisions of $\mathbf{E}$, where $\mathbf{E}_j \in \mathcal{A}$. This reformulation enables the modeling of the entire embedding space evolution $\mathbf{E} \to \hat{\mathbf{E}}$ as the evolution of a set of subspaces $\{\mathbf{E}_j \to \hat{\mathbf{E}}_j\}_{j=1}^K$. In practice, we define each subspace $\mathbf{E}_j$ as the embedding representation of a mini-batch $\mathcal{D}_B$. In subsequent evaluations, we find that different batch sampling strategies do not significantly affect model convergence under mainstream protocols.

Since explicitly modeling $z$ requires solving an intractable inverse problem in a high-dimensional space, and $z$ is coupled with $\mathbf{E}_j$ as part of the posterior condition in $q_\psi(\hat{\mathbf{E}}|\mathbf{E}, z)$, we introduce a conditioning mapping $f(\cdot; \mathbf{W}_z)$ with learnable parameters $\mathbf{W}_z$ to embed the latent representation $z$ into each batch of pre-trained embeddings. Based on this, the posterior is transformed into $q_\psi(\hat{\mathbf{E}}_j|\boldsymbol{\Theta}_j)$, where $\boldsymbol{\Theta}_j = f(\mathbf{E}_j; \mathbf{W}_z)$, enabling end-to-end optimization of the integration of $z$ through the evolution of each subspace.

**Pseudo-cluster center generation.** Given the posterior conditioning $\boldsymbol{\Theta}_j$, the final state $\hat{\mathbf{E}}_j$ is required to learn the evolution mapping $\psi : \boldsymbol{\Theta}_j \to \hat{\mathbf{E}}_j$. However, obtaining the true $\hat{\mathbf{E}}_j$ is impractical, as it requires full fine-tuning of the model on the downstream task, which is computationally unaffordable. To address this limitation, we leverage findings that modern pre-trained models exhibit strong convergence capabilities on training data [36, 39], resulting in well-separated distributions in the embedding space after convergence. Specifically, representations of different classes tend to form distinct clusters, reflecting the static structure of the final embedding space.

Based on these inferences, we adopt a pseudo-cluster center generation strategy to mimic the static distribution separability of the evolved embedding space. The pseudo-cluster centers $\{\mathbf{c}_i\}_{i=1}^C$ are designed to ensure a well-separated structure in the final embedding space. They can be generated using one-hot vectors, random vectors sampled from high-dimensional spaces, or eigenvectors obtained through Principal Component Analysis (PCA) or Laplacian-based methods to promote sparser distributions. Additionally, a shifting strategy based on the statistical properties $(\mu, \sigma)$ of the initial feature distribution $\mathbf{E}$ is applied to further accelerate model convergence. Finally, we utilize the pseudo-cluster centers as the final state $\hat{\mathbf{E}} = \{\mathbf{c}_y \mid y \in Y\}$ for subsequent estimation. A detailed comparison of clustering generation strategies is provided in the appendix.

**Deparametric approximation.** After dividing the embedding space and setting the final state, our goal is to learn the mapping $\psi : \mathbf{\Theta}_j \rightarrow \hat{\mathbf{E}}_j$, between the synthesized condition $\mathbf{\Theta}_j$ and the final state $\hat{\mathbf{E}}_j$ to estimate the embedding space after fine-tuning. A straightforward approach would involve optimizing $\mathbf{W}_g$ **separately within each mini-batch** through iterative updates using the function $g(\mathbf{\Theta}_j; \mathbf{W}_g)$. However, performing such per-batch optimization across all subspaces during training and validation would incur prohibitively high computational costs, and is also difficult to implement within existing frameworks.

To address this, we propose a deparametric approximation of the embedding space evolution, leveraging a dynamic equation-based formulation to eliminate the need for explicit optimization of $\mathbf{W}_g$. This allows efficient transferability estimation while preserving the underlying adaptation dynamics.

Specifically, we consider a mapping layer $g$ parameterized by $\mathbf{W}_g$, which is used to model the posterior distribution of $\hat{\mathbf{E}}$ by transforming the synthesized condition $\mathbf{\Theta}_j$ into its final state $\hat{\mathbf{E}}_j$. This transformation at iteration $n$ is expressed as $\mathbf{E}_j^{(n)} = \mathbf{\Theta}_j \mathbf{W}_g^{(n)}$, where $n$ denotes the $n$-th update step of $\mathbf{W}_g$. As a representative example, considering the MSE loss as $\mathcal{L}_{pc}$, the loss function over a mini-batch of size $B$ can be formulated as:

$$
\begin{aligned}
\mathcal{L}_{pc}(\mathbf{W}_g^{(n)}) &= \frac{1}{2B}||\mathbf{E}_j^{(n)} - \hat{\mathbf{E}}_j||_2^2 \\
&= \frac{1}{B}\text{Tr}((\mathbf{\Theta}_j \mathbf{W}_g^{(n)} - \hat{\mathbf{E}}_j)^T(\mathbf{\Theta}_j \mathbf{W}_g^{(n)} - \hat{\mathbf{E}}_j)).
\end{aligned}
\tag{2}
$$

Thus, the gradient descent update rule for $\mathbf{W}_g$ becomes:

$$
\begin{cases}
\frac{\partial \mathcal{L}_{pc}}{\partial \mathbf{W}_g^{(n)}} = \frac{1}{B}[\mathbf{\Theta}_j^T \mathbf{\Theta}_j \mathbf{W}_g^{(n)} - \mathbf{\Theta}_j^T \hat{\mathbf{E}}_j] \\
\mathbf{W}_g^{(n+1)} = \mathbf{W}_g^{(n)} - \frac{\eta}{B} \cdot \frac{\partial \mathcal{L}_{pc}}{\partial \mathbf{W}_g^{(n)}},
\end{cases}
\tag{3}
$$

where $\eta$ denotes the learning rate. The corresponding evolution of the subspace $\mathbf{E}_j^{(n)}$ follows:

$$
\begin{aligned}
\mathbf{E}_j^{(n+1)} &= \mathbf{\Theta}_j \mathbf{W}_g^{(n)} - \frac{\eta}{B} \cdot \mathbf{\Theta}_j \frac{\partial \mathcal{L}_{pc}}{\partial \mathbf{W}_g^{(n)}} \\
&= (\mathbf{I} - \frac{\eta}{B}\mathbf{\Theta}_j \mathbf{\Theta}_j^T)\mathbf{\Theta}_j \mathbf{W}_g^{(n)} + \frac{\eta}{B}\mathbf{\Theta}_j \mathbf{\Theta}_j^T \hat{\mathbf{E}}_j.
\end{aligned}
\tag{4}
$$

where we define the constant matrix $\mathbf{C} = \frac{1}{B}\mathbf{\Theta}_j \mathbf{\Theta}_j^T$, which depends only on the initial input condition $\mathbf{\Theta}_j$. Thus, the update can be rewritten as:

$$
\mathbf{E}_j^{(n+1)} = (\mathbf{I} - \eta\mathbf{C})\mathbf{E}_j^{(n)} + \eta\mathbf{C}\hat{\mathbf{E}}_j.
\tag{5}
$$

This deparametric approximation removes the dependency on the learnable parameters $\mathbf{W}_g$ and enables transferability estimation without iterative updates for each subspace, significantly reducing computational overhead while preserving estimation accuracy.

### 3.4 Framework

Building on the proposed Divide-and-Conquer Variational Approximation (DVA), the full ITM framework is constructed as illustrated in Fig. 2. During estimation, the latent transferability $z$ is implicitly embedded into each subset of the pre-trained embeddings $\mathbf{E}_j$ to form $\mathbf{\Theta}_j$ via $f(\cdot; \mathbf{W}_z)$. Then, the deparametric approximation update is performed based on the synthesized posterior condition $\mathbf{\Theta}_j$ and the pseudo-cluster $\hat{\mathbf{E}}$ for the specific downstream task.

To maintain compatibility with diverse downstream tasks, we incorporate the downstream objective $\mathcal{L}_{obj}$ into ITM's optimization, as illustrated in Fig. 2(c). This ensures the learned embedding evolution to align with task-specific requirements. The objective $\mathcal{L}_{obj}$ typically takes the form of cross-entropy loss for standard classification tasks, but alternative formulations can be applied as needed for other scenarios, making ITM broadly applicable beyond classical classification and enhancing its generalization potential. During evaluation, we assess the accuracy of the evolved embedding spaces on the evaluation set $\mathcal{D}_{\mathcal{E}}$ as the estimated score. The training procedure is detailed in the appendix.

# 4 Experiments

## 4.1 Benchmark

Following prior work [17, 4, 7, 6], we construct a benchmark based on 10 classic single-label image classification datasets. However, with the rise of foundation visual models and improved training protocols, existing benchmarks may not fully capture model capacities. For comprehensive comparison between ITM and state-of-the-art methods, we select 10 recent pre-trained vision models and fine-tune them using integrated and enhanced protocols. The downstream datasets, pre-trained model zoo, and fine-tuning procedures are detailed below. By integrating the models from prior benchmarks, we develop an extended benchmark of 20 models, which is detailed in the appendix.

**Downstream datasets.** We use 10 widely adopted single-label image classification datasets for transfer learning, primarily sourced through the official PyTorch [40] (e.g., torchvision.datasets): CIFAR-10, CIFAR-100 [41], FGVC Aircraft [42], Caltech-101 [43], DTD [44], Oxford-IIIT Pets [45], Stanford Cars [46], SUN-397 [47], Food-101 [48], and Oxford 102 Flowers [49]. These datasets span diverse scenarios, with varying class counts and dataset sizes, offering a comprehensive evaluation of the generalization ability of vision foundation models.

**Pre-trained model zoo.** Prior TE methods largely focus on models with similar architectures and supervised pre-training, limiting generalization to newer models adopting diverse architectures and self-supervised strategies. To cover common downstream use cases, we select 10 representative models across supervised, contrastive, and masked image modeling (MIM) pre-training.

Specifically, we include ResNet-18 [26], MobileNetV2 [50], EfficientNet-B0 [51], and DenseNet-121 [27] for supervised learning; DINO-S8 [10], DINO-B16 [10], and MoCov3-B16 [9] for contrastive learning; and MAE-B16 [11], MAE-L16 [11], and SimMIM-B16 [12] for MIM. Here, S/B/L denote small, base, and large ViT [8] models, while 8/16 refer to the patch size.

**Fine-tuning protocols.** Fine-tuned performance is critical for accurate TE ranking. However, settings used in prior work [17, 4, 7, 6] often underexploit modern models, introducing evaluation bias. To address this, we follow official implementations [10, 51, 26, 11, 12] and adopt AdamW [52] to jointly fine-tune backbones and classification heads for 100 epochs. Learning rates are grid-searched over $10^{-5}, 2 \times 10^{-5}, 5 \times 10^{-5}$ and weight decays over $10^{-2}, 10^{-4}$. Evaluation is performed every epoch, and the best checkpoint is used for ground-truth ranking. All experiments use a single NVIDIA V100 GPU (32 GB), with batch sizes of 64 for classification and 8 for segmentation. Further fine-tuning details and performance comparisons are provided in the appendix.

## 4.2 Implementation details

To balance accuracy and efficiency, we set the ITM training iterations to 500. The transferability score $s$ is evaluated every 100 iterations, with the highest score recorded as the final estimation for each model. During training, 4/5 of the official training split from each benchmark is randomly sampled for optimizing the ITM framework, while the remaining 1/5 is reserved for score calculation. The step size $\eta$ in DVA is fixed at 0.01, and the iteration count $n$ is determined adaptively (details provided in the appendix). Across all experiments, the learning rate $\alpha$ is set to $5 \times 10^{-3}$, and AdamW [52] is adopted as the optimizer. All comparisons are conducted under a consistent environment with 8-core CPUs to ensure fairness. Pre-trained models, benchmarks, and code will be publicly released to facilitate reproduction and future research. In the efficiency comparison, we exclude the basic time of feature extraction, following previous works [7, 6], to provide a clean comparison of the running time of the TE process.

## 4.3 Comparison with previous approaches

**Benchmark comparison.** We evaluate the proposed ITM against state-of-the-art methods on the benchmark, with results summarized in Tab. 1.

As shown in Tab. 1, ITM significantly outperforms all counterparts in weighted Kendall's $\tau_w$, achieving a substantial performance margin. While existing methods struggle to generalize to newer architectures due to the growing diversity of model collections and increasingly complex convergence behaviors, ITM leverages implicit transferability modeling to deliver more accurate estimation and stronger generalization capabilities. Specifically, ITM achieves the highest average

Table 1: **Comparison of weighted Kendall's $\tau_w$ and wall-clock time (in seconds) across different methods on the benchmark datasets. Each wall-clock time does not include the average feature extraction time of 738 seconds (on GPU), but only corresponds to the running time of the TE method (on CPU).**

| Methods | Cal101 | Cars | CIFAR100 | CIFAR10 | DTD | Aircraft | Flowers | Food | Pets | SUN | Avg. |
|---|---|---|---|---|---|---|---|---|---|---|---|
| | | | | Weighted Kendall's $\tau_w$ ↑ | | | | | | | |
| $\mathcal{N}$Leep [25] | 0.47 | 0.04 | 0.32 | 0.48 | 0.57 | 0.13 | 0.62 | 0.24 | 0.30 | 0.01 | 0.32 |
| LogME [4] | **0.71** | 0.36 | 0.56 | 0.61 | 0.61 | 0.22 | **0.77** | 0.15 | 0.14 | 0.38 | 0.45 |
| PARC [18] | 0.08 | 0.00 | -0.07 | 0.25 | 0.42 | 0.12 | 0.62 | 0.19 | 0.10 | 0.01 | 0.17 |
| SFDA [17] | 0.59 | 0.07 | 0.48 | **0.79** | 0.13 | 0.18 | -0.39 | 0.33 | 0.28 | 0.09 | 0.25 |
| ETran [5] | 0.13 | -0.06 | -0.14 | 0.21 | 0.36 | 0.27 | 0.08 | 0.23 | 0.38 | -0.06 | 0.14 |
| PED [7] | 0.32 | -0.01 | 0.51 | 0.77 | 0.06 | -0.20 | 0.16 | **0.60** | -0.20 | 0.07 | 0.21 |
| SA (LDA) [22] | 0.31 | -0.11 | -0.06 | 0.34 | 0.33 | 0.22 | 0.14 | 0.18 | 0.33 | -0.12 | 0.16 |
| **ITM (Ours)** | 0.56 | **0.61** | **0.59** | 0.69 | **0.77** | **0.43** | 0.65 | 0.44 | **0.73** | **0.62** | **0.61** |
| | | | | Wall-clock Time (s) ↓ | | | | | | | |
| $\mathcal{N}$Leep [25] | 25.52 | 44.91 | 862.49 | 268.25 | 4.60 | 17.65 | 3.42 | 1387.65 | 4.31 | 47.33 | 266.61 |
| LogME [4] | 0.85 | 1.32 | 4.50 | 2.62 | 0.50 | 0.86 | 0.54 | 6.29 | 0.53 | 1.31 | 1.93 |
| PARC [18] | 14.42 | 19.82 | 116.65 | 118.19 | 0.74 | 13.14 | 0.25 | 106.19 | 3.53 | 19.80 | 41.27 |
| SFDA [17] | 4.02 | 5.43 | 20.11 | 18.56 | 1.70 | 3.92 | 1.91 | 28.86 | 2.20 | 5.43 | 9.21 |
| ETran [5] | 1.71 | 2.30 | 7.58 | 6.88 | 0.77 | 1.64 | 0.98 | 10.63 | 0.96 | 2.31 | 3.58 |
| PED [7] | 6.99 | 8.31 | 34.32 | 46.80 | 2.33 | 5.95 | 2.80 | 41.72 | 3.94 | 8.29 | 16.14 |
| SA (LDA) [22] | 4.65 | 6.08 | 10.29 | 9.04 | 2.92 | 4.48 | 4.11 | 13.82 | 3.16 | 6.08 | 6.46 |
| **ITM (Ours)** | 7.50 | 8.50 | 9.40 | 7.90 | 7.00 | 7.50 | 7.20 | 10.50 | 7.20 | 11.50 | 8.42 |

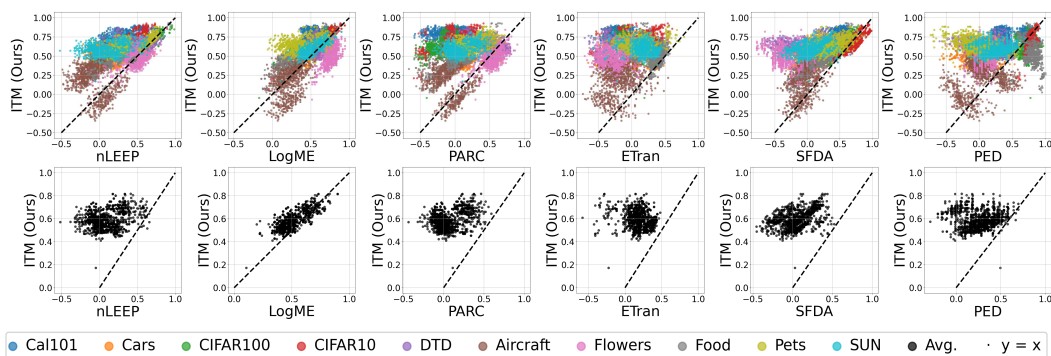

Figure 3: **Comparison of stability between ITM and state-of-the-art methods.** The axes show the weighted Kendall's $\tau_w$ scores of each method. Points appearing above the $y = x$ dashed line indicate cases where ITM achieves higher estimation accuracy than its counterparts.

weighted Kendall's $\tau_w$ (0.61 vs. 0.45), demonstrating clear improvements over previous state-of-the-art methods and validating its effectiveness. We further rerun ITM with five different random seeds, resulting in a score of $0.60 \pm 0.01$, reflecting its stability and robustness across varying initializations.

In terms of running time, ITM remains highly competitive. It requires only 8.42 seconds on average to evaluate across the benchmark, which is negligible compared to the 738 seconds needed to compute the initial embedding spaces shared by all TE methods. Although slightly slower than LogME and ETran, ITM outperforms both by a large margin in accuracy (0.61 vs. 0.45 and 0.14), achieving a better balance between accuracy and efficiency.

**Stability evaluation.** To further evaluate ITM's stability, we conduct additional experiments following the methodology of [21]. We include four classical CNN models, as in prior studies [7, 17], to expand the benchmark and better assess generalization across architectures. In each trial, 10 models are randomly sampled from a pool of 14, and weighted Kendall's $\tau_w$ is computed to evaluate the performance of each TE method. This results in 1001 combinations ($C_{14}^{10}$), providing a robust measure of stability. Fig. 3 compares ITM against existing approaches.

As shown in Fig. 3, ITM consistently outperforms other TE methods in a wide range of experiments, demonstrating superior stability in the selection of random subsets of models. These results underscore the strong generalizability and robustness of ITM.

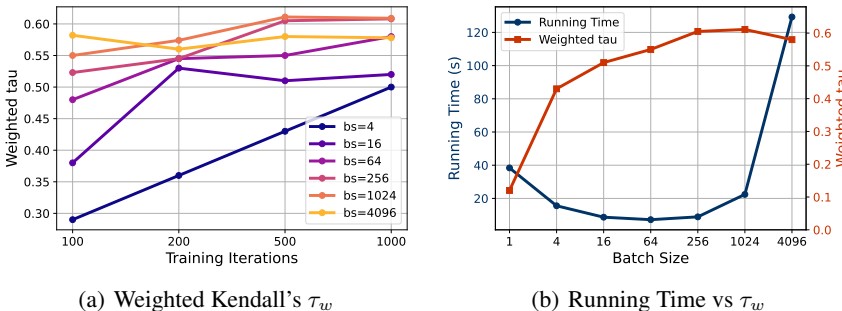

(a) Weighted Kendall's $\tau_w$           (b) Running Time vs $\tau_w$

Figure 4: **Ablation study on batch size.** (a) Weighted Kendall's $\tau_w$ across different combinations of batch size and iteration count. (b) Impact of batch size on weighted Kendall's $\tau_w$ and running time.

Table 2: **Quantitative ablation study evaluating different candidate loss functions for $\mathcal{L}_{\mathbf{pc}}$.** Losses include Mean Absolute Error (MAE), Cross-Entropy (CE), and Mean Squared Error (MSE).

| $\mathcal{L}_{pc}$ | Cal101 | Cars | CIFAR100 | CIFAR10 | DTD | Aircraft | Flowers | Food | Pets | SUN | $\tau_w$ |
|---|---|---|---|---|---|---|---|---|---|---|---|
| CE | 0.717 | 0.483 | 0.573 | 0.689 | 0.602 | 0.323 | 0.582 | 0.437 | 0.619 | 0.519 | 0.554 |
| MAE | 0.717 | 0.458 | 0.590 | 0.689 | 0.629 | 0.377 | 0.575 | 0.437 | 0.714 | 0.470 | 0.566 |
| **MSE** | 0.560 | 0.608 | 0.590 | 0.689 | 0.768 | 0.428 | 0.651 | 0.437 | 0.734 | 0.616 | **0.608** |

## 4.4 Ablation study

**Batch division.** Batch-wise division controls the approximation granularity in DVA's posterior decomposition and is critical to ITM's performance. We study the impact of batch size and iteration count on weighted Kendall's $\tau_w$ and running time, as shown in Fig. 4.

From Fig. 4, we observe that as batch size increases from 1 to 1024, accuracy improves consistently, since larger batches better estimate transferability. However, further increasing batch size to 4096 slightly degrades accuracy while significantly increasing running time, likely because an overly large batch overwhelms DVA and hampers convergence. Balancing efficiency and accuracy, we set batch size to 256 and iteration count to 500 in the final configuration.

**Losses in DA.** During the deparametric approximation, we utilize $\mathcal{L}_{pc}$ for pseudo-clustering-based updating. To fully assess the choice of loss functions, we compare classical losses including MSE, MAE, and CE. The results are shown in Tab. 2.

As observed in Tab. 2, MSE achieves the highest average correlation ($\tau_w = 0.608$), outperforming MAE ($\tau_w = 0.566$) and CE ($\tau_w = 0.554$). Although results vary slightly across individual datasets, the overall trend suggests that MSE provides the most robust estimation of transferability. This may be attributed to the smoothing properties of MSE, which lead to more stable pseudo-cluster updates during the approximation process. Based on this, we adopt MSE as the default loss function for $\mathcal{L}_{pc}$.

**Task generalization.** Since ITM eliminates dependence on single-label supervision during estimation, it offers practical potential to generalize beyond image classification tasks. Unlike previous counterparts that are constrained by task-specific supervision and architectural coupling, ITM can be readily extended to dense prediction tasks such as semantic segmentation, which require additional task heads and fine-grained feature modeling. To validate this capability, we construct a proxy benchmark for segmentation tasks, including five widely used foundation models evaluated on the CamVid [53] and Cityscapes [54] datasets. The results are presented in Tab. 3.

As shown in Tab. 3, ITM delivers stable and accurate transferability predictions even for segmentation tasks, successfully selecting the most suitable models for downstream segmentation applications. This demonstrates the strong generalization ability and practical applicability of ITM. In contrast, existing methods such as PED, LogME, and SFDA—designed around single image-level classification labels—struggle to generalize effectively to new tasks.

Table 3: **Quantitative ablation study on semantic segmentation.** We conduct experiments on two dense prediction datasets, CamVid [53] and Cityscapes [54], using five pre-trained ViT models.

| Datasets | Metrics | Models | | | | | $\tau_w$ |
|---|---|---|---|---|---|---|---|
| | | MoCov3-B16 | DINO-B16 | MAE-B16 | SimMIM-B16 | MAE-L16 | |
| CamVid | mIoU | 58.11 | 60.05 | 63.99 | 64.52 | **68.25** | 0.61 |
| | ITM score | 85.87 | 86.41 | 88.58 | 83.85 | **89.03** | |
| Cityscapes | mIoU | 40.06 | 41.45 | 44.21 | 43.72 | **47.33** | 0.72 |
| | ITM score | 79.77 | 79.14 | 83.11 | 78.03 | **83.86** | |

## 5   Conclusion

We present Implicit Transferability Modeling (ITM), a framework for transferability estimation that models embedding space evolution during fine-tuning as a posterior distribution, with latent integration capturing transferability. To enhance scalability, we introduce a Divide-and-Conquer Variational Approximation (DVA), enabling efficient embedding evolution without explicit fine-tuning. Experiments on an enhanced benchmark across diverse architectures and pre-training strategies show that ITM consistently outperforms state-of-the-art methods in effectiveness and generalization.

**Limitations.** While ITM demonstrates stronger generalization than prior methods, it remains constrained by its reliance on embedding space discrimination and cannot yet directly handle complex supervision scenarios such as detection or vision-language tasks. A common limitation of current TE methods is their dependence on final output features $\mathbf{E} = \phi(\mathbf{X})$, often overlooking the intrinsic properties of the model $\phi$. Although ITM partially addresses this via latent integration, more fine-grained modeling—such as leveraging intermediate representations or enriched output embeddings—may be necessary to further improve transferability estimation. From another perspective, most existing TE methods focus primarily on performance under full fine-tuning. However, the properties of base models within the rapidly evolving PEFT [55, 56, 57, 58] paradigm may not align with such assumptions. It remains an open challenge to extend transferability estimation frameworks to accurately predict performance under parameter-efficient adaptation schemes.

## Acknowledgment

This work is partly supported by the National Key Research and Development Plan (2024YFB3309302), National Natural Science Foundation of China (82441024), the Beijing Natural Science Foundation (L251073), the Research Program of State Key Laboratory of Complex and Critical Software Environment, and the Fundamental Research Funds for the Central Universities.

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

# A    Technical Appendices and Supplementary Material

## A.1    Radar Chart

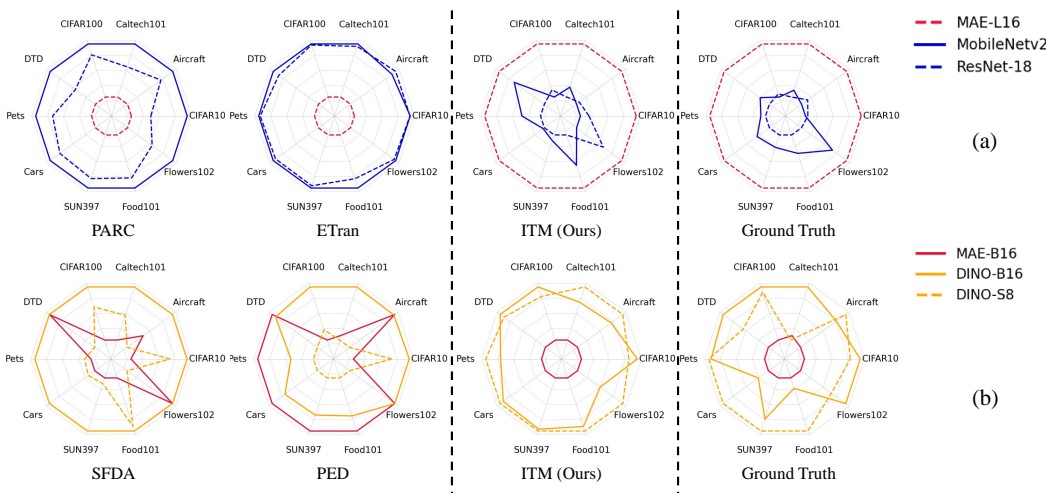

Figure 5: **Comparison between ITM, recent TE methods, and ground-truth performance across ten benchmarks.** Scores are normalized to $[0.3, 1]$ for clearer visualization. (a) Evaluation on MAE and CNN pre-trained models. Static TE methods (e.g., PARC [18] and ETran [5]) fail to generalize to MAE [11] due to weak discriminative power in initial embeddings. (b) Evaluation on DINO and MAE pre-trained models. Dynamic TE methods (e.g., SFDA [17] and PED [7]) overestimate MAE's performance. In contrast, ITM generalizes well across diverse pre-trained models and provides accurate estimations.

## A.2    Pseudo-code

---
**Algorithm 1** Training process of DVA without deparametric approximation
---
**Require:** Embedding of training dataset $\mathbf{E}$ extracted by model $\phi$, target cluster center $\hat{\mathbf{E}}$, target label $\mathbf{Y}$, linear layers $f, g, h$, number of training iterations $T$, learning rate $\alpha$, number of subspace iterations $M$ and learning rate of subspace $\eta$
**Ensure:** Optimized parameters $\mathbf{W}_z, \mathbf{W}_g, \mathbf{W}_h$ of linear layers $f, g, h$.
  1: Randomly initialize $\mathbf{W}_z, \mathbf{W}_h$
  2: **for** $t = 1$ to $T$ **do**
  3:      Sample one batch of data: $(\mathbf{E}_t, \hat{\mathbf{E}}_t, \mathbf{Y}_t) \leftarrow \text{sample}(\mathbf{E}, \hat{\mathbf{E}}, \mathbf{Y}, t)$
  4:      Compute $\mathbf{\Theta}_t$: $\mathbf{\Theta}_t \leftarrow f(\mathbf{E}_t; \mathbf{W}_z)$
  5:      Randomly initialize $\mathbf{W}_g$
  6:      **for** $m = 1$ to $M$ **do**
  7:         Compute gradient: $grad_g \leftarrow \nabla_{\mathbf{W}_g} \text{MSE}(g(\mathbf{\Theta}_t), \hat{\mathbf{E}}_t)$
  8:         Update parameters of $g$: $\mathbf{W}_g \leftarrow \mathbf{W}_g - \eta \cdot grad_g$
  9:      **end for**
10:      Compute logits: $logits \leftarrow h(g(\mathbf{\Theta}_t))$
11:      Compute gradient: $grad_f \leftarrow \nabla_{\mathbf{W}_z} \text{CE}(logits, \mathbf{Y}_t)$, $grad_h \leftarrow \nabla_{\mathbf{W}_h} \text{CE}(logits, \mathbf{Y}_t)$
12:      Update parameters of $f, h$: $\mathbf{W}_z \leftarrow \mathbf{W}_z - \alpha \cdot grad_f$, $\mathbf{W}_h \leftarrow \mathbf{W}_h - \alpha \cdot grad_h$
13: **end for**
14: **return** $W_z, W_g, W_h$
---

As shown in Algorithm 1, DVA requires multiple training steps for the parameter $W_g$ during each macro iteration, followed by updates to the parameters $W_z$ and $W_h$. This "training within training" approach is not only challenging to implement but also inefficient. However, it clearly demonstrates the core idea of DVA: training the batch-specific parameter $W_g$ exclusively for each data batch.

**Algorithm 2** Training process of DVA

---

**Require:** Embedding of training dataset $\mathbf{E}$ extracted by model $\phi$, target cluster center $\hat{\mathbf{E}}$, target label $\mathbf{Y}$, linear layers $f, h$, number of training iterations $T$, learning rate $\alpha$, number of subspace iterations $n$ and learning rate of subspace $\eta$

**Ensure:** Optimized parameters $\mathbf{W}_z, \mathbf{W}_h$ of linear layers $f, h$.

1: Randomly initialize $\mathbf{W}_z, \mathbf{W}_h$
2: **for** $t = 1$ to $T$ **do**
3:      Sample one batch of data: $(\mathbf{E}_t, \hat{\mathbf{E}}_t, \mathbf{Y}_t) \leftarrow \text{sample}(\mathbf{E}, \hat{\mathbf{E}}, \mathbf{Y}, t)$
4:      Compute $\mathbf{\Theta}_t$: $\mathbf{\Theta}_t \leftarrow \mathbf{W}_z \mathbf{E}_t$
5:      deparametric approximation: $\mathbf{E}_t^{(n)} \leftarrow (I - \eta \mathbf{C})^n (\mathbf{\Theta}_t - \hat{\mathbf{E}}_j) + \hat{\mathbf{E}}_j$ (Eq. 5)
6:      Compute logits: $logits \leftarrow h(\mathbf{E}_t^{(n)})$
7:      Compute gradient: $grad_f \leftarrow \nabla_{\mathbf{W}_z} \text{CE}(logits, \mathbf{Y}_t), grad_h \leftarrow \nabla_{\mathbf{W}_h} \text{CE}(logits, \mathbf{Y}_t)$
8:      Update parameters of $f, h$: $\mathbf{W}_z \leftarrow \mathbf{W}_z - \alpha \cdot grad_f, \mathbf{W}_h \leftarrow \mathbf{W}_h - \alpha \cdot grad_h$
9: **end for**
10: **return** $W_z, W_h$

---

As shown in Algorithm 2, we introduced the deparametric approximation method, which eliminates the need for training the linear layer $g$ in lines 5 to 9 of Algorithm 1. Instead, the feature output $\mathbf{E}_t^{(n)}$ after several iterations is directly obtained using $n$ and $\eta$. Additionally, we employ the Pseudo-clustering Optimization method to generate pseudo-cluster centers that guide the convergence direction of $\mathbf{E}_t^{(n)}$. Finally, $f$ and $h$ are trained directly using CrossEntropy loss function.

### A.3  Role of Latent Variable $z$

Table 4: Cosine similarity matrix for the learned mapping parameters $W_z$. The background colors highlight the two main patterns of consistency. **Red cells** show high intra-task similarity. **Yellow cells** show medium intra-model similarity.

|      | D-A  | M-A  | C-A  | D-F  | M-F  | C-F  | D-R  | M-R  | C-R  |
|------|------|------|------|------|------|------|------|------|------|
| D-A  | **1.00** | 0.197 | 0.181 | 0.133 | 0.133 | 0.133 | 0.138 | 0.134 | 0.133 |
| M-A  | 0.197 | **1.00** | 0.215 | 0.133 | 0.142 | 0.134 | 0.131 | 0.158 | 0.134 |
| C-A  | 0.181 | 0.215 | **1.00** | 0.131 | 0.132 | 0.135 | 0.132 | 0.132 | 0.142 |
| D-F  | 0.133 | 0.133 | 0.131 | **1.00** | 0.340 | 0.318 | 0.137 | 0.133 | 0.134 |
| M-F  | 0.133 | 0.142 | 0.132 | 0.340 | **1.00** | 0.349 | 0.132 | 0.152 | 0.136 |
| C-F  | 0.133 | 0.134 | 0.135 | 0.318 | 0.349 | **1.00** | 0.132 | 0.133 | 0.139 |
| D-R  | 0.138 | 0.131 | 0.132 | 0.137 | 0.132 | 0.132 | **1.00** | 0.301 | 0.283 |
| M-R  | 0.134 | 0.158 | 0.132 | 0.133 | 0.152 | 0.133 | 0.301 | **1.00** | 0.309 |
| C-R  | 0.133 | 0.134 | 0.142 | 0.134 | 0.136 | 0.139 | 0.283 | 0.309 | **1.00** |

ITM is designed to estimate transferability at the level of model-task interaction, rather than treating the model or task independently. The latent variable $z$ is introduced to capture this interaction between a model's representational capacity and the semantic demands of a downstream task. This task conditioned nature of $z$ is essential, as the effectiveness of a pretrained model depends not only on its own features but also on their alignment with the target task.

Since explicitly modeling $z$ is intractable due to the high-dimensional and non-linear nature of embedding evolution (and the absence of supervision for transferability), ITM adopts an implicit modeling strategy. For each model-task pair, a mapping function $f_t(\cdot | W_z)$, parameterized by $W_z$, is learned. This function is shared across all subspaces (batches) for the pair and operates on conditioned embeddings that encode both the static properties of the pretrained model $\phi_d$ and the distributional characteristics of the downstream task data. Combined with the pseudo-cluster-based update trajectory, this function simulates how representations adapt to a new task, without requiring end-to-end fine-tuning.

To further validate and interpret the learned latent variable $z$, we analyze the mapping parameters $W_z$ across different model-task pairs. We perform Singular Value Decomposition (SVD) on each $W_z$ and compare the left singular vectors using cosine similarity. These vectors represent the dominant transformation directions in the input space and provide a principled basis for comparing the learned functions. We study three pretrained models–DINO-B16 (D), MAE-B16 (M), and MoCov3-B16 (C)–on three tasks: Aircraft (A), Flowers (F), and Cars (R). The resulting similarity matrix is shown in Tab. 4. From this analysis, we observe three salient patterns:

- **Task Consistency:** Within each task (e.g., Aircraft, Flowers, Cars), the intra-task similarity among different models (i.e., the 3×3 diagonal blocks) remains consistently high. This suggests that ITM learns task-specific transformation functions that are stable across diverse backbones, highlighting its ability to capture task-dependent adaptation behavior.
- **Model Consistency:** For each model (e.g., DINO-B16), the inter-task similarity—i.e., similarity between the same model across different tasks—is noticeably higher than that of unrelated model-task pairs. This implies that ITM effectively encodes model-specific inductive biases into the learned mappings.
- **Task Dominance:** Overall, task-driven consistency is stronger than model-driven consistency, indicating the downstream task has a greater influence than the model choice. This aligns with Table 4, where performance variance is more pronounced across tasks than across models.

## A.4 Benchmark

Table 5: **Accuracy of different models across datasets under optimized fine-tuning settings.**

| Models | Cal101 | Cars | CIFAR100 | CIFAR10 | DTD | Aircraft | Flowers | Food | Pets | SUN | VOC |
|---|---|---|---|---|---|---|---|---|---|---|---|
| DenseNet-121 | 97.23 | 88.39 | 85.67 | 97.38 | 69.47 | 83.86 | 90.41 | 85.00 | 91.77 | 72.49 | 82.75 |
| DenseNet-161 | 98.21 | 89.84 | 87.01 | 97.79 | 72.61 | 88.24 | 92.00 | 86.72 | 93.24 | 74.10 | 82.27 |
| DenseNet-169 | 97.29 | 88.83 | 86.24 | 97.75 | 70.90 | 83.80 | 91.58 | 85.58 | 92.91 | 73.76 | 81.85 |
| DenseNet-201 | 97.47 | 89.55 | 86.39 | 97.65 | 72.98 | 84.04 | 91.07 | 86.14 | 93.08 | 74.13 | 80.86 |
| EfficientNet-B0 | 97.87 | 87.25 | 87.01 | 97.88 | 68.88 | 81.61 | 89.71 | 85.82 | 90.68 | 73.87 | 81.10 |
| GoogleNet | 96.54 | 86.54 | 83.39 | 96.97 | 69.73 | 83.29 | 88.01 | 81.36 | 90.60 | 69.67 | 80.45 |
| InceptionV3 | 96.95 | 84.73 | 83.42 | 96.92 | 66.75 | 79.60 | 87.14 | 81.79 | 92.86 | 68.07 | 79.50 |
| MnasNet | 95.56 | 81.12 | 83.40 | 96.78 | 60.96 | 75.01 | 76.48 | 82.87 | 90.65 | 70.02 | 78.78 |
| MobileNetv2 | 96.03 | 85.65 | 82.15 | 96.48 | 67.34 | 76.36 | 90.03 | 83.69 | 89.02 | 71.16 | 82.45 |
| ResNet-18 | 95.74 | 83.58 | 82.69 | 96.57 | 65.59 | 77.56 | 88.52 | 79.92 | 88.50 | 68.71 | 76.74 |
| ResNet-34 | 96.54 | 86.11 | 85.31 | 97.30 | 67.50 | 77.53 | 90.03 | 85.24 | 92.70 | 70.75 | 81.56 |
| ResNet-50 | 97.58 | 88.87 | 84.60 | 97.70 | 70.85 | 84.49 | 91.38 | 87.08 | 93.65 | 74.85 | 84.19 |
| ResNet-101 | 97.47 | 88.71 | 87.58 | 98.14 | 71.17 | 85.75 | 90.91 | 87.97 | 94.00 | 77.21 | 83.83 |
| ResNet-152 | 97.64 | 89.07 | 88.58 | 98.01 | 71.44 | 81.64 | 89.25 | 88.35 | 94.79 | 76.18 | 84.73 |
| DINO-B16 | 98.16 | 88.42 | 90.53 | 98.70 | 74.31 | 78.40 | 93.19 | 87.88 | 92.97 | 77.02 | 84.51 |
| DINO-S8 | 97.47 | 89.33 | 90.24 | 98.65 | 71.97 | 80.02 | 90.84 | 90.84 | 93.10 | 77.53 | 85.58 |
| MoCov3-B16 | 97.70 | 88.53 | 90.74 | 98.68 | 72.07 | 75.70 | 93.61 | 87.15 | 89.75 | 75.92 | 80.37 |
| MAE-B16 | 97.52 | 88.16 | 87.55 | 98.42 | 69.04 | 72.16 | 85.80 | 87.30 | 89.81 | 75.29 | 82.86 |
| MAE-L16 | 97.98 | 91.21 | 91.28 | 98.55 | 74.26 | 85.30 | 90.73 | 90.82 | 94.69 | 79.18 | 88.01 |
| SimMIM-B16 | 96.20 | 86.02 | 88.80 | 98.63 | 66.17 | 68.32 | 83.87 | 87.88 | 87.16 | 74.55 | 79.62 |

As shown in Tab. 5 and Tab. 6, we present the performance of twenty different pre-trained models on eleven downstream single-label classification datasets. Unlike previous works such as PED [7] and SFDA [17], we adopt a more practical training setup by using the superior AdamW optimizer and training for 100 epochs. This results in consistently better model performance in Tab. 5 compared to prior methods, as shown in Fig. 6. The improvement is particularly evident on larger datasets such as CIFAR-[10,100], SUN397, and Food101, where previous approaches like PED, which only train models for 5k iterations, which is insufficient to fully capture the true performance of the models.

As shown in Tab. 7, we adopt consistent training split configurations across all datasets. Standard accuracy serves as the metric for ground-truth ranking computation. All datasets are sourced from the official torchvision.datasets module of PyTorch [40].

## A.5 Other evaluation protocols

We evaluated the transferability estimation methods based on other rank correlation metrics, including Spearman's $\rho$ [59] and Kendall's $\tau$ [60]. As shown in Tab. 8, ITM also achieved the best performance on these two metrics.

Table 6: **Comparison of benchmark attributes in terms of model diversity and fine-tuning protocols.** "Mix" indicates the inclusion of models from multiple categories for evaluating TE methods.

| TE methods | Model structures | | | Pre-train methods | | | | Full-finetune settings | | |
|---|---|---|---|---|---|---|---|---|---|---|
| | CNN | ViT | Mix | Supervised | Contrastive learning | Masked image modeling | Mix | Batch size | Optimizer | Iterations |
| SFDA, SA, PED, LEAD | ✓ | | | ✓ | ✓ | | | 64 | SGD | 5000 iterations (Avg. 16.2 epochs) |
| ITM (Ours) | ✓ | ✓ | ✓ | ✓ | ✓ | ✓ | ✓ | 64 | AdamW | 100 epochs |

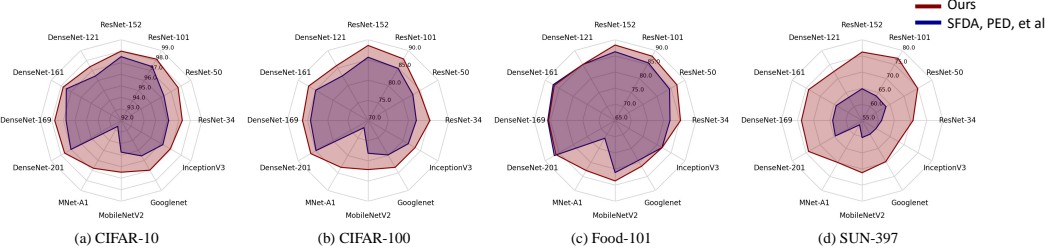

(a) CIFAR-10     (b) CIFAR-100     (c) Food-101     (d) SUN-397

Figure 6: **Performance comparison of model zoo entries across benchmarks.** Under identical dataset settings across four benchmarks, the benchmark adopted in ITM better realizes the potential of each model and consistently delivers stronger performance than those used in prior studies, providing a more realistic evaluation of real-world model selection scenarios.

Table 7: **Comparison of dataset settings across benchmarks.** "Trainval" denotes joint use of training and validation sets. "mCA" indicates mean per-class accuracy, while "Acc" refers to overall accuracy. ITM adopts standard splits and a unified evaluation metric for fair benchmarking.

| settings | Methods | Cal101 | Cars | CIFAR100 | CIFAR10 | DTD | Aircraft | Flowers | Food | Pets | SUN | VOC |
|---|---|---|---|---|---|---|---|---|---|---|---|---|
| Training dataset | SFDA, PED, *et al.* | train | train | train | train | trainval | trainval | trainval | train | train | train | trainval |
| | ITM (Ours) | train | train | train | train | train | train | train | train | train | train | train |
| Testing metric | SFDA, PED, *et al.* | mCA | Acc | Acc | Acc | Acc | mCA | mCA | Acc | mCA | Acc | mCA |
| | ITM (Ours) | Acc | Acc | Acc | Acc | Acc | Acc | Acc | Acc | Acc | Acc | Acc |

Table 8: **Comparison of Spearman's $\rho$ [59] and Kendall's $\tau$ [60] for different methods on various datasets.**

| Methods | Cal101 | Cars | CIFAR100 | CIFAR10 | DTD | Aircraft | Flowers | Food | Pets | SUN | Avg. |
|---|---|---|---|---|---|---|---|---|---|---|---|
| | | | | | Spearman's $\rho$ | | | | | | |
| PED | 0.61 | 0.05 | 0.81 | 0.92 | 0.19 | -0.37 | -0.07 | 0.84 | -0.18 | 0.25 | 0.31 |
| LogME | 0.81 | 0.61 | 0.62 | 0.61 | 0.77 | 0.45 | 0.90 | 0.20 | 0.37 | 0.52 | 0.59 |
| $\mathcal{N}$Leep | 0.37 | 0.24 | 0.25 | 0.42 | 0.53 | 0.44 | 0.79 | 0.09 | 0.41 | 0.01 | 0.35 |
| PARC | -0.01 | 0.10 | -0.19 | 0.05 | 0.49 | 0.43 | 0.79 | 0.03 | 0.26 | -0.05 | 0.19 |
| SFDA | 0.71 | 0.16 | 0.68 | 0.70 | 0.09 | 0.38 | -0.58 | 0.43 | 0.52 | 0.13 | 0.32 |
| ETran | -0.01 | -0.08 | -0.28 | 0.01 | 0.09 | 0.54 | 0.13 | 0.05 | 0.39 | -0.13 | 0.07 |
| **ITM (Ours)** | 0.75 | 0.84 | 0.71 | 0.70 | 0.87 | 0.54 | 0.78 | 0.45 | 0.84 | 0.75 | **0.72** |
| | | | | | Kendall's $\tau$ | | | | | | |
| PED | 0.556 | 0.067 | 0.644 | 0.778 | 0.111 | -0.333 | -0.061 | 0.689 | -0.156 | 0.111 | 0.241 |
| LogME | 0.600 | 0.422 | 0.422 | 0.467 | 0.600 | 0.333 | 0.778 | 0.111 | 0.244 | 0.378 | 0.436 |
| $\mathcal{N}$Leep | 0.289 | 0.156 | 0.244 | 0.289 | 0.422 | 0.289 | 0.600 | 0.156 | 0.333 | 0.067 | 0.284 |
| PARC | 0.022 | 0.111 | -0.156 | 0.022 | 0.378 | 0.289 | 0.600 | 0.067 | 0.200 | 0.067 | 0.160 |
| SFDA | 0.511 | 0.156 | 0.467 | 0.600 | 0.067 | 0.244 | -0.479 | 0.244 | 0.422 | 0.111 | 0.234 |
| ETran | -0.022 | -0.022 | -0.156 | -0.022 | 0.111 | 0.422 | 0.111 | 0.156 | 0.378 | -0.067 | 0.089 |
| **ITM (Ours)** | 0.629 | 0.644 | 0.556 | 0.600 | 0.719 | 0.422 | 0.600 | 0.378 | 0.719 | 0.600 | **0.587** |

## A.6 Benchmark of CNN models

Following the setup of SFDA [17], we also establish a benchmark on 14 supervised pre-trained CNN models, including DenseNet-[121,161,169,201] [27], ResNet-[34,50,101,152] [26], InceptionV3 [61], GoogleNet [62], MobileNetV2 [50], and MnasNet [63]. As shown in Tab. 9, our method continues to achieve the best performance on traditional benchmarks. Additionally, when ranking supervised pre-trained CNN models, most previous methods still perform better even as the number of models increases to 12, compared to their performance when handling a more diverse set of models (MIM, ID). This further highlights the limitations of existing TE methods in the effective handling of diverse pre-trained models.

Table 9: **Comparison of weighted Kendall's $\tau_w$ [60] for different methods on CNN benchmark of SFDA [17].**

| Methods | Cal101 | Cars | CIFAR100 | CIFAR10 | DTD | Flowers | Food | Pets | SUN | VOC | Avg. |
|---|---|---|---|---|---|---|---|---|---|---|---|
| PED | 0.401 | 0.230 | 0.559 | 0.625 | -0.113 | 0.190 | 0.511 | 0.750 | -0.101 | 0.446 | 0.350 |
| LogME | 0.558 | 0.402 | 0.750 | 0.761 | 0.340 | 0.141 | 0.780 | 0.664 | 0.817 | 0.758 | 0.597 |
| $\mathcal{N}$Leep | 0.586 | 0.373 | 0.814 | 0.656 | 0.100 | -0.013 | 0.776 | 0.475 | 0.736 | 0.593 | 0.509 |
| PARC | 0.093 | 0.352 | -0.456 | -0.553 | 0.033 | -0.134 | -0.595 | 0.516 | 0.676 | -0.119 | -0.019 |
| SFDA | 0.501 | 0.485 | 0.791 | 0.761 | -0.236 | 0.232 | 0.312 | 0.797 | 0.099 | 0.619 | 0.436 |
| ETran | -0.301 | -0.223 | -0.414 | 0.043 | -0.114 | -0.159 | -0.363 | -0.560 | -0.458 | -0.227 | -0.278 |
| SA (+LogME) | 0.703 | 0.251 | 0.738 | 0.761 | 0.470 | 0.074 | 0.872 | 0.676 | 0.570 | 0.811 | 0.593 |
| **ITM (Ours)** | 0.478 | 0.572 | 0.805 | 0.739 | 0.535 | 0.445 | 0.832 | 0.681 | 0.370 | 0.781 | **0.624** |

## A.7 Adaptive $n$

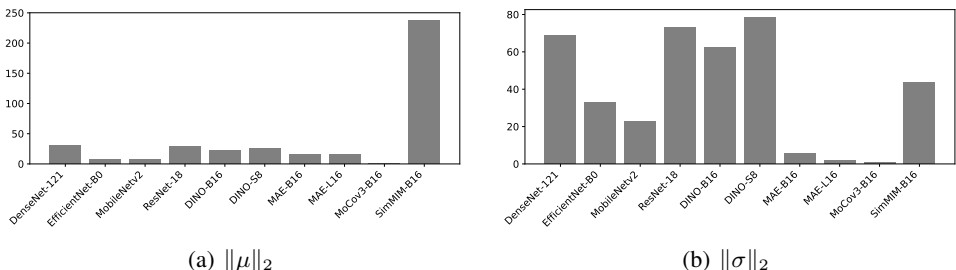

(a) $\|\mu\|_2$         (b) $\|\sigma\|_2$

Figure 7: **Visualization of the divergence in initial embedding distributions across different pre-trained models on Caltech-101.** The mean and variance are computed to highlight differences in the static embedding states.

The goal of DVA is to map the embedding spaces of the pre-trained backbone before and after full fine-tuning. However, as shown in Fig. 7, different pre-trained models have initial embedding spaces with different means and variances, which means that using the same $\eta$ in equation 5 may be unfair to models with initially poor feature distributions. Since equation 5 is similar to a linear recurrence, we computed the average Euclidean distance of all initial features in initial embedding space $\mathbf{E} = \{f_i\}_i^n$ of model $\phi$, after standardization, to the cluster centers of their respective classes: $\mathrm{dis}_\phi = \frac{1}{n}\sum_{j=1}^n \|f_i' - \bar{f}_{y_i}'\|_2$ where $f_i' = \frac{f_i - \bar{f}}{\sigma(f)}$ represents the standardized features, and $\bar{f}_c'$ denotes the mean of the features for class $y_i = c$.

Since the linear recurrence coefficient $\eta C = \frac{\eta}{B}\boldsymbol{\Theta}_j\boldsymbol{\Theta}_j^T$ is complex, we consider the recurrence relation when batch size $B = 1$ and $\eta = \eta_0$:

$$\mathbf{E}_i^{(n_i)} = (1 - \eta_0)^{n_i}\boldsymbol{\Theta}_i + (1 - (1 - \eta_0)^{n_i})\hat{\mathbf{E}}_i, \tag{6}$$

$$\mathbf{E}_j^{(n_j)} = (1 - \eta_0)^{n_j}\boldsymbol{\Theta}_j + (1 - (1 - \eta_0)^{n_j})\hat{\mathbf{E}}_j. \tag{7}$$

This represents two initial spaces $\boldsymbol{\Theta}_i$ and $\boldsymbol{\Theta}_j$ being updated towards the target $\hat{\mathbf{E}}_i$ and $\hat{\mathbf{E}}_j$ using different numbers of iterations $n_i$ and $n_j$. By setting $\|\mathbf{E}_i^{(n)} - \hat{\mathbf{E}}_i\|_2 = \|\mathbf{E}_j^{(n)} - \hat{\mathbf{E}}_j\|_2$, we obtain the equation:

$$\|(1 - \eta_0)^{n_i}\boldsymbol{\Theta}_i + (1 - (1 - \eta_0)^{n_i})\hat{\mathbf{E}}_i - \hat{\mathbf{E}}_i\|_2 = \|(1 - \eta_0)^{n_j}\boldsymbol{\Theta}_j + (1 - (1 - \eta_0)^{n_j})\hat{\mathbf{E}}_j - \hat{\mathbf{E}}_j\|_2$$
$$\implies (1 - \eta_0)^{n_i}\|\boldsymbol{\Theta}_i - \hat{\mathbf{E}}_i\|_2 = (1 - \eta_0)^{n_j}\|\boldsymbol{\Theta}_j - \hat{\mathbf{E}}_j\|_2. \tag{8}$$

This implies that $(1 - \eta_0)^n$ is inversely proportional to the Euclidean distance between the features and the target point. Therefore, we compute an adaptive learning rate $\eta$ based on the model's $\mathrm{dis}_\phi$:

$$n_i = f(\mathrm{dis}_{\phi_i}) = \lceil \log_{1-\eta_0} \frac{\mathrm{dis}_b}{\mathrm{dis}_{\phi_i}} \rceil + n_b \tag{9}$$

In Eq. (9) we set $n_b = 20, \eta_0 = 0.01$ as an anchor for the model $\phi$ when $\mathrm{dis}_b = 1.0$.

In practice, for batch sizes $B > 1$, the effectiveness of the aforementioned derivation is primarily restricted to the component of $\boldsymbol{\Theta}$ parallel to the pseudo-cluster center $\hat{\mathbf{E}}$. Combined with the effective

$1/B$ scaling of the learning rate $\eta_0$, this causes the adaptive $n$ mechanism to gradually degenerate towards a fixed $n_b$ with increasing $B$. As a result, the ITM approach can still use adaptive $n$ with large batch sizes while preserving stability (as shown in Fig. 4), though its capacity to balance the convergence speeds across different models will be reduced. Furthermore, as the batch size increases, the convergence of ITM itself also accelerates. This diminishes the necessity for adaptive $n$ to provide nuanced adjustments to update step lengths (or to fine-tune the update progression), making it possible to achieve comparable performance even with a fixed $n_b$.

### A.8 Pseudo-cluster centers

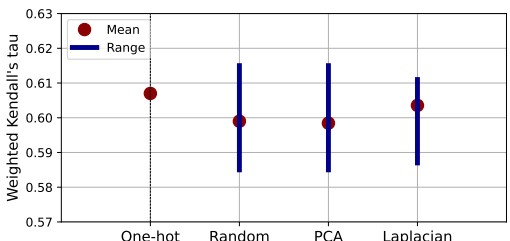

Figure 8: **Performance of different methods for generating orthogonal pseudo-cluster centers.** Mean and variance are computed over 10 independent runs to assess robustness and consistency.

As shown in Fig. 8, we explored different methods for generating standard orthogonal pseudo-cluster centers on the unit sphere as targets for feature alignment. Specifically, "PCA" and "Laplacian" refer to orthogonalization techniques applied after randomly generating centers using Principal Component Analysis (PCA) and the Laplacian matrix, respectively. Through repeated experiments, we observed that the performance across these methods exhibited minimal variation. This can be attributed to the fact that the classification head $h$ applies an additional linear transformation to the embedded features, making the choice of pseudo-cluster centers robust. Furthermore, it can be mathematically proven that two sets of standard orthogonal bases $\mathbf{P}$ and $\mathbf{Q}$ can be transformed by an orthogonal matrix $\mathbf{W}$, such that $\mathbf{P} = \mathbf{W}\mathbf{Q}$. In this scenario, the distance distribution of any feature in the space relative to these basis pseudo-cluster centers remains invariant before and after the transformation. Thus, even a simple linear classification head is robust to different standard orthogonal pseudo-cluster centers. To address the issue of random instability during practical use, we adopted one-hot encoding of the classes as the cluster centers.

### A.9 Comparison with mutual information

While ITM does not explicitly compute mutual information (MI), its estimated scores can be viewed as empirical proxies for model–task alignment, which relates to—but goes beyond—MI. Conventional MI captures statistical overlap between datasets, but fails to account for representational alignment or task-specific dynamics. ITM instead estimates how well a model's features can adapt to a target task via latent, model-conditioned evolution—capturing both semantic and structural compatibility. Thus, ITM offers a more fine-grained and dynamic perspective than traditional MI-based metrics.

### A.10 Task generalization

The successful application to semantic segmentation highlights ITM's broader potential. In principle, ITM is task-agnostic as its core DVA framework models the evolution of the embedding space, which is not inherently tied to a specific output type. The framework's use of a general downstream objective, $\mathcal{L}_{obj}$, allows it to be adapted to any discriminative task where a task-specific head can be applied to the evolved embeddings. This makes ITM a valuable tool for a wide range of applications beyond classification and segmentation. For instance, in complex video analysis tasks such as video semantuc segmentation [64, 65], a critical first step is often the selection of a powerful pre-trained image model to act as a frame-level feature extractor or backbone. ITM provides a practical and efficient solution for this crucial model selection process, enabling researchers to identify the most suitable backbone without extensive experimentation for use in subsequent, more complex or specialized downstream tasks.

## A.11 Embedding space mapping

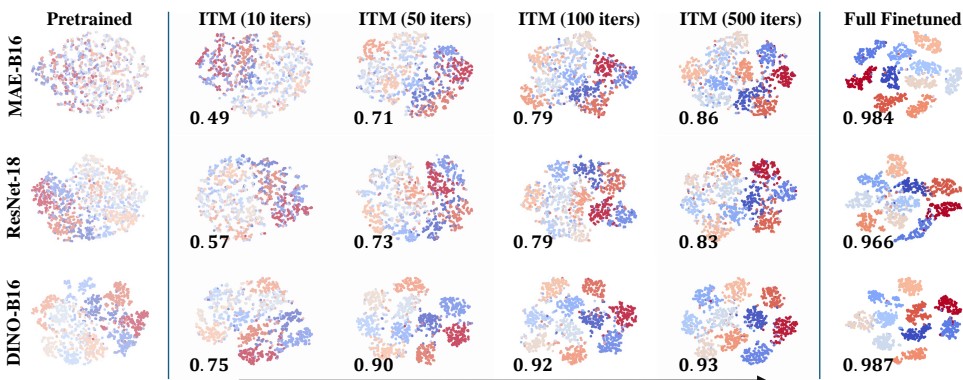

Figure 9: **Illustration of the embedding evolution process.** T-SNE visualizations of the embedding distributions for ResNet-18, DINO-B16, and MAE-B16 on the CIFAR-10 test set, showing both initial and final states, along with intermediate states $\mathbf{E}_j^{(n)}$ during ITM evolution. The number in the lower-left corner of each plot indicates the model's test accuracy at that stage.

As shown in Fig. 9, the embedding distributions of pre-trained models $\mathbf{E}$ from different pre-training methods are quite disparate, but after fine-tuning, their embedding space $\hat{\mathbf{E}}$ converges to a much better distribution. Previous methods either solely rely on pre-trained model features or use crude methods to simulate the fine-tuning process, which explains why these methods perform poorly on such complex benchmarks.

## A.12 Licenses

The CNN models and datasets used in this work leverage the PyTorch framework and the official torchvision datasets, both licensed under the 3-Clause BSD License. We gratefully acknowledge the contributions of the PyTorch development team. The licensing information for other models is shown in the Tab. 10.

Table 10: **Licenses of ViT models.**

|  | MoCov3 | DINO | MAE | SimMIM |
|---|---|---|---|---|
| License | CC BY-NC 4.0 | Apache-2.0 | CC BY-NC 4.0 | MIT |

