# OpenReview forum: "Implicit Modeling for Transferability Estimation of Vision Foundation Models"
_NeurIPS.cc/2025/Conference — NeurIPS 2025 poster_

### Official Review · Reviewer_zA6C · 2025-06-29

**Clarity:** 3
**Significance:** 2
**Originality:** 3
**Rating:** 4
**Confidence:** 3

**Summary:**

This paper proposes Implicit Transferability Modeling (ITM), a framework for estimating how well a pretrained vision model will perform on a new downstream task, without full fine-tuning. Unlike prior methods that explicitly simulate embedding space evolution during adaptation, ITM introduces a latent variable representing model transferability and a Divide-and-Conquer Variational Approximation (DVA) to efficiently simulate embedding adaptation in subspaces. Experiments are conducted on on a range of downstream tasks and models, showing performance improvement in both accuracy and efficiency.

**Questions:**

Please refer to the weaknesses section above

**Ethical Concerns:**

["NO or VERY MINOR ethics concerns only"]

**Final Justification:**

I thank the authors for their detailed response. Their rebuttal has addressed most of my concerns, so I will increase my score to 4.

**Quality:**

2

**Strengths And Weaknesses:**

**Strengths**

- The paper proposes a novel idea: learn a latent representation of transferability instead of hardcoding how models evolve. This makes the method more flexible and scalable.
- The method clearly improves over past work on standard datasets. It runs faster than many dynamic baselines while also being more accurate.

**Weaknesses**

- The DVA, may be an oversimplification of the actual fine-tuning process. The deparametric approximation is derived from a simple MSE loss objective, resulting in a linear iterative update toward fixed pseudo-cluster centers. Real fine-tuning is a complex, non-linear process, typically driven by cross-entropy loss and sophisticated optimizers like AdamW. It is questionable whether the proposed linear evolution simulation can realistically capture these dynamics. The framework may not be modeling fine-tuning so much as it is measuring how well a model's initial embeddings can be linearly mapped to separated clusters—a useful proxy, perhaps, but a proxy nonetheless.

- The central claim of "implicitly modeling transferability" using `z` is ambiguous. The framework learns a single set of weights $W_z$ on a subset of the downstream task data to create a conditioned embedding $Θ_j$. This seems to capture properties of the task rather than the intrinsic, pre-trained properties of a specific model. The framework doesn't appear to explicitly model anything unique about $\phi_i$ and $\phi_j$ other than their initial feature distributions $E$.

- The latent variable z is introduced as a representation of “intrinsic transferability,” but it is neither grounded nor validated.
There is no effort to interpret it, probe its consistency across tasks, or show whether it captures architectural, data-driven, or task-alignment priors.

- The framework's reported indifference to the geometry of the target pseudo-cluster centers is concerning. This suggests the method is capturing a very coarse signal about class separability, which is only useful upto classification and not fine-grained/dense tasks.

- How is the independence of the K subspaces ensured? It is unclear how realistic and well enforced this assumption is

 - The method assumes that the final fine-tuned embedding space is cluster-separable, and approximates this via pseudo-centers. But this assumption is quite strong and fragile. It breaks down on tasks where class boundaries are not well-separated in the embedding space (e.g., fine-grained visual categorization, heavy class imbalance, or noisy labels). Nearly all considered tasks are clean image classification datasets.

- Figure 2 is extremely dense and hard to parse.

- Overall, the paper is well written and clear. However, it is hard to follow at a few places. For example, Section 3 is quite dense and a lot of details (such as Deparametric approximation) are only briefly explained. A more detailed explanation would really help enhance clarity.

---

> ### Author Rebuttal · Authors · 2025-07-31
>
> We thank the reviewer for your recognition of our method's novelty and the supieror performance. Below, we address each point in detail and clarify key motivations and design decisions in ITM.
>
> **[W1:Simplicity of DVA and Linear Updates]** While full fine-tuning is a complex, non-linear process involving sophisticated losses and optimizers, ITM is not designed to replicate this process in its entirety. Rather, its goal is to approximate the overall trajectory of representational evolution in a lightweight and tractable manner—sufficient to estimate relative model-task compatibility without requiring expensive fine-tuning.
>
> The proposed DVA framework serves as a practical surrogate. It approximates embedding evolution through batch-wise updates toward pseudo-cluster centers, but crucially, this process is guided by a learnable function $f(\cdot \mid W_z)$, optimized across the entire embedding space. This enables ITM to capture task-conditioned adaptation patterns unique to each model-task pair, without resorting to static or hand-crafted transformations.
>
> As a result, ITM’s latent evolution is significantly more expressive than static projection-based baselines or separability heuristics in prior work. We will clarify this motivation and modeling strategy more explicitly in the final version.
>
> **[W2:Role of Latent Variable z in Modeling Transferability]** ITM is designed to estimate transferability at the level of model-task interaction, rather than treating the model or task independently. The latent variable $z$ is introduced to capture this interaction between a model’s representational capacity and the semantic demands of a downstream task. This task-conditioned nature of $z$ is essential, as the effectiveness of a pretrained model depends not only on its own features but also on their alignment with the target task.
>
> Since explicitly modeling $z$ is intractable due to the high-dimensional and non-linear nature of embedding evolution (and the absence of supervision for transferability), ITM adopts an implicit modeling strategy. For each model-task pair, a mapping function $f(\cdot \mid W_z)$, parameterized by $W_z$, is learned. This function is shared across all subspaces (batches) for the pair and operates on conditioned embeddings that encode both the static properties of the pretrained model $\phi_i$ and the distributional characteristics of the downstream task data.
>
> Combined with the pseudo-cluster-based update trajectory, this function simulates how representations adapt to a new task, without requiring end-to-end fine-tuning. We will make the role of $z$ and the modeling mechanism more intuitive in the final version.
>
> **[W3: Additional Evaluation on Latent z]** We thank the reviewer for the insightful suggestion. To further validate and interpret the learned latent variable $z$, we analyze the mapping parameters $W_z$ across different model-task pairs.
>
> We perform Singular Value Decomposition (SVD) on each $W_z$ and compare the left singular vectors using cosine similarity. These vectors represent the dominant transformation directions in the input space and provide a principled basis for comparing the learned functions.
>
> We study three pretrained models—DINO-B16 (D), MAE-B16 (M), and MoCov3-B16 (C)—on three tasks: Aircraft (A), Flowers (F), and Cars (R). The resulting similarity matrix is:
>
> | |D-A|M-A|C-A|D-F|M-F|C-F|D-R|M-R|C-R|
> |-|-|-|-|-|-|-|-|-|-|
> |**D-A**|**1.00**|**0.197**|**0.181**|0.133|0.133|0.133|`0.138`|0.134|0.133|
> |**M-A**|**0.197**|**1.00**|**0.215**|0.133|`0.142`|0.134|0.131|`0.158`|0.134|
> |**C-A**|**0.181**|**0.215**|**1.00**|0.131|0.132|`0.135`|0.132|0.132|`0.142`|
> |**D-F**|0.133|0.133|0.131|**1.00**|**0.340**|**0.318**|`0.137`|0.133|0.134|
> |**M-F**|0.133|`0.142`|0.132|**0.340**|**1.00**|**0.349**|0.132|`0.152`|`0.136`|
> |**C-F**|0.133|0.134|`0.135`|**0.318**|**0.349**|**1.00**|0.132|0.133|`0.139`|
> |**D-R**|`0.138`|0.131|0.132|`0.137`|0.132|0.132|**1.00**|**0.301**|**0.283**|
> |**M-R**|0.134|`0.158`|0.132|0.133|`0.152`|0.133|**0.301**|**1.00**|**0.309**|
> |**C-R**|0.133|0.134|`0.142`|0.134|`0.136`|`0.139`|**0.283**|**0.309**|**1.00**|
>
> From this analysis, we observe three salient patterns:
> - Task Consistency: Within each task (e.g., Aircraft, Flowers, Cars), the intra-task similarity among different models (i.e., the 3×3 diagonal blocks) remains consistently high. This suggests that ITM learns task-specific transformation functions that are stable across diverse backbones, highlighting its ability to capture task-dependent adaptation behavior.
> - Model Consistency: For each model (e.g., DINO-B16), the inter-task similarity—i.e., similarity between the same model across different tasks—is noticeably higher than that of unrelated model–task pairs. This implies that ITM effectively encodes model-specific inductive biases into the learned mappings.
> - Task Dominance: Overall, task-driven consistency is stronger than model-driven consistency, indicating the downstream task has a greater influence than the model choice. This aligns with Table 4, where performance variance is more pronounced across tasks than across models.
>
> **[W4:Geometry of Pseudo-Cluster Centers]** We appreciate the reviewer’s concern. ITM does not rely on the precise geometry of the pseudo-cluster centers. These centers serve as flexible soft anchors to guide the batch-wise representational updates—not as rigid or semantically grounded class prototypes. Their purpose is to shape the direction of representation evolution without requiring assumptions such as spherical clusters or fixed inter-class margins.
>
> Moreover, the downstream classifier $h$ is independently learned and can adapt to the evolved embedding distribution, even when class boundaries are non-linear or overlapping. This allows ITM to generalize well to tasks that go beyond simple class separability.
>
> We demonstrate this empirically in Table 3 and Appendix A.7, including segmentation tasks where dense spatial predictions are required. The consistent transferability estimation across both classification and dense prediction tasks supports that ITM is not limited by coarse geometric assumptions.
>
> **[W5: Assumption of Independent K Subspaces]** We would like to clarify that ITM does not strictly enforce subspace independence. Instead, this is a practical approximation adopted to enable scalable and localized modeling. Within the DVA framework, each batch is treated as a locally stable manifold—a heuristic motivated by prior work in contrastive and manifold learning, which has shown that local neighborhoods in the embedding space tend to exhibit semantic coherence and reduced entanglement.
>
> To evaluate the robustness of this design choice, we compare three batch sampling strategies (random division, uniform sampling, and max-variance sampling) on CIFAR-10. The table below summarizes the maximum variation (Δ) in ITM scores across strategies at different iteration steps:
>
> |Iteration Step|MAE-B16|ResNet-18|DINO-B16|
> |-|-|-|-|
> |200|0.014|0.004|0.001|
> |600|0.006|0.011|0.004|
> |1000|0.005|0.003|0.002|
>
> These consistently low variations demonstrate that ITM's estimates are highly stable and robust across different subspace configurations, confirming that the subspace assumption does not impact estimation reliability.
>
> We will include this analysis in the final version and revise Section 3 to more clearly describe the rationale and implications of the subspace decomposition.
>
> **[W6:Cluster-Separability Assumption]** As noted earlier, ITM does not assume hard or perfect separability of the fine-tuned embedding space. The pseudo-cluster centers in our framework are not rigid class prototypes, but soft attractors that guide representation evolution. We make no assumptions about spherical clusters, uniform inter-class distances, or clean label boundaries—this is by design to ensure flexibility and robustness in diverse, imperfect real-world scenarios.
>
> To support this, we evaluate ITM on 13 downstream tasks that include both classification and segmentation, covering a wide range of complexity levels, label noise, and class distributions. These datasets include fine-grained recognition tasks, imbalanced categories, and dense prediction benchmarks, which present substantial spatial and structural variability. Across these challenging scenarios, ITM consistently demonstrates strong correlation with actual transfer performance (see Table 1, Table 3, and reply to ZajB-W1), suggesting that it does not rely on strict cluster separability to be effective.
>
> It is also important to distinguish between the pseudo-centers used during optimization and the actual structure of the evolved embeddings. The pseudo-centers are heuristically initialized targets that serve as anchors for the iterative DVA process. However, the final embedding organization is not defined by these initial centers, but rather shaped dynamically through interaction between the model and task during training. As shown in Figure 9, the resulting embedding clusters reflect the underlying adaptation process rather than enforcing a pre-specified geometry.
>
> We will revise the final version to clarify this conceptual distinction and emphasize that ITM does not rely on strong separability assumptions.
>
> **[W7: Figure2 and Section3- Clarity and Detail ]** Thank you for the suggestion. We will revise Figure 2 by separating the three main stages into clearer sequential panels with better visual flow. Additionally, we will expand Section 3.3 with a more detailed explanation of the deparametric approximation, including its motivation and design choices, to improve clarity and reader understanding.
>
> We are committed to enhancing the clarity and accessibility of the final version. Please feel free to reach out with any remaining questions—we are happy to provide further clarification.

---

### Official Review · Reviewer_Ly76 · 2025-06-30

**Clarity:** 1
**Significance:** 2
**Originality:** 2
**Rating:** 4
**Confidence:** 3

**Summary:**

This paper presents an approach to build a transferability score between a pretrained model and a target dataset.

This is done by extracting latent features by running the model on the dataset, and then learning a mapping between the latent space and the final fine-tuning performance. The innovation consist of splitting the dataset into batches and to process sub-region of the latent space corresponding to these batches independently.

I should emphasize that I thought the paper very hard to follow and I am not sure of my understanding, despite being used to the literature about transfer learning.

**Questions:**

See questions in the weaknesses section.

**Ethical Concerns:**

["NO or VERY MINOR ethics concerns only"]

**Final Justification:**

After rebuttal

**Limitations:**

yes

**Quality:**

2

**Strengths And Weaknesses:**

**Strengths**:
- The results obtained by the proposed method appear to be good on the tested datasets and pretrained models. They often outperform the approaches used for comparison.


**Weaknesses**:
- I thought the paper was really hard to follow and I struggled to understand what is being done, and what motivates the choices made by the authors. Many concepts, ideas and notations are introduced without clear explanations and justifications. For example:
    - What is the rationale behind splitting the space into batches? It is central to the proposed approach, and I could not find a clear justification for this methodological choice.
    - What is the scope of the learnable function to map embeddings to transferability? Is there a different function for each pretrained model? For each subspace of each pretrained model? Is it the same function (shared weights) for all pretrained models?

- Experiments are not sufficient:
    - Datasets: only classification datasets, why not experimenting with more complex datasets (segmentation, detection)? The claims talks about downstream tasks in the broad sense, but the experiments only present classification. This limited scope should be clearly stated in the title, abstract, intro. Later on, we discover that the approach was tested on segmentation, why was it not mentioned in the section about datasets description? Why not comparing the different approaches on the semantic segmentation datasets?
    - Models: 10 pretrained models is really not a lot. Model zoos have way over 1000 models and the real challenge today is how we select a good model among this large pool of models.
    - Wall-clock time appears particularly fast. I find it hard to believe that it takes less than 10 seconds to assess the transferability of 10 models on CIFAR100, knowing that we need at least one forward pass of each model on each of the 50k images. More details are needed to understand what exactly is presented here.
    - Comparsion to the literature: Missing references to the compared approaches and details about why they were selected.
    - “In contrast, existing methods such as PED, LogME, and SFDA—designed around single image-level classification labels—struggle to generalize effectively to new tasks.” —> I don’t see what supports this claim in Table 3.

- In the introduction, I found the limitations of the existing literature very abstract. The result is that I couldn’t grasp the gaps that this paper attempts to fill. For example, “existing TE techniques are primarily evaluated on models with similar architectures and training paradigms […] and fail to generalize to models trained with advanced pre-training strategies or novel architectural designs” —> I don’t see why the approaches presented just above are architecture-dependent or training paradigm-dependent. More explanations are required. Another example: “previous studies attempt to model these dynamics by directly simulating the evolution of embedding spaces, they fail to comprehensively capture both aspects, thereby limiting their generalization capabilities.” —> This is very abstract, what aspects? Why did previous attempts fail? How did they fail? When I reached line 40, it was still not clear what limitations the paper is trying to overcome.



Note: My concerns about this paper are mostly due to the fact that I believe the paper needs heavy rewriting to improve clarity. I might be wrong in my assessment and if other reviewers feel that the paper is understandable, feel free to favor their review over mine.

---

> ### Author Rebuttal · Authors · 2025-07-31
>
> Thank you for your thoughtful and detailed feedback. We sincerely appreciate your effort in reviewing the paper. Below, we provide point-by-point responses and clarifications.
>
> **[Summary: Clarity and Readability]**
> We appreciate your comments and regret that parts of the manuscript may have been unclear. While other reviewers found the paper well-organized, we acknowledge that the introduction may lack sufficient conceptual grounding for readers unfamiliar with *transferability estimation* (TE), an emerging but distinct field from traditional transfer learning.
>
> TE focuses on predicting how well a pretrained model will perform on a new task *without fine-tuning*, avoiding costly trial-and-error across large model pools. Unlike transfer learning, which adapts model weights, TE produces a *score* for guiding efficient model selection.
>
> Whereas existing TE approaches rely on *static heuristics* or *rule-based dynamics*, our method introduces a principled framework that models *latent representation evolution*. This enables more accurate, generalizable TE across diverse architectures, pretraining paradigms, and—for the first time—complex tasks like semantic segmentation.
>
> We will revise the introduction and related work sections to clarify these goals and contributions, helping make the paper more accessible without compromising rigor.
>
> **[W1-1 Motivation for Batch-Wise Space Division]**
> Prior TE methods typically model the global embedding space, which is high-dimensional, entangled, and often dependent on specific architectures—limiting generalization.
>
> In contrast, ITM adopts a divide-and-conquer approach by modeling local representation dynamics in batch-wise partitions of the embedding space. This allows ITM to capture consistent and fine-grained adaptation patterns while maintaining scalability.
>
> We will revise the manuscript to highlight this motivation more clearly and explain its theoretical and practical benefits.
>
> **[W1-2 Scope of the Learnable Mapping Function]**
> To assist your understanding, here is a consensus in the existing field of transferability estimation: Given a model zoo $\{\phi_i\}$ and a series of downstream datasets $\{D_i\}$, all TE methods require that for each $<\phi_p, D_q>$ pair, the features of all samples are extracted to form an initial Embedding $E_{pq}$. A prediction score is then given by a TE function, $TE(E_{pq}, Y_q)$, where $Y_q$ represents the labels for task $D_q$. In other words, for $M$ models and $N$ tasks, the TE method must run $M \cdot N$ inferences for feature extraction and $M \cdot N$ TE function calls for calculating the TE score. Therefore, for our ITM, all the method descriptions are based on the $E$ and $Y$ obtained from a single pair $<\phi, D>$.
>
> In the transferability estimation (TE) literature, the core objective of the learnable mapping function is to predict how well a given pretrained model will adapt to a target task—typically by approximating how its feature representations would evolve post fine-tuning. Depending on the method design, this mapping may be shared across models or conditioned on each model-task pair.
>
> For example, static TE methods often adopt a shared function to directly map from initial embeddings to separability scores, while dynamic methods may attempt to simulate adaptation trajectories in a model- or task-specific manner (see Sec. 2 for a taxonomy).
>
> ITM follows this general paradigm but introduces a more discriminative and adaptive formulation. Specifically, ITM learns a unique mapping function for each model–task pair, parameterized by a latent variable $\mathbf{W}_z$, which governs a lightweight evolution function operating at the batch level. This function takes a local batch of embeddings as input and outputs their evolved representations, effectively approximating the dynamics of fine-tuning without requiring access to gradient-based training.
>
> This localized and modular design enables ITM to efficiently capture model–task-specific adaptation signals while remaining computationally lightweight and broadly applicable across architectures and tasks. Importantly, it avoids brute-force simulation or end-to-end optimization, distinguishing ITM from prior approaches.
>
> We will revise Section 3 in the final version to clarify the scope and benefits of this mapping function design.
>
> **[W2-1 Dataset Scope and Presentation]**
> We fully agree that broader evaluation across diverse and complex downstream tasks is crucial for advancing transferability estimation (TE). In fact, one of the key contributions of our work is the explicit extension of TE beyond standard classification to more complex settings such as semantic segmentation.
>
> While prior TE methods—including NLEEP, LogME, PARC, SFDA, ETran, PED, and LEAD—have focused almost entirely on single-label image classification and CNN-based backbones, our approach is the first to support dense prediction tasks. Not only did we construct a more authoritative benchmark with 20 models and 11 image classification datasets (as shown in Figure 6), but we also tested ITM's ability to generalize to the semantic segmentation task on pre-trained ViT models. To our knowledge, this is the most diverse benchmark to date in the TE literature, covering both CNN and ViT backbones, and a range of pretraining strategies such as supervised learning, contrastive learning (e.g., MoCo, DINO), and masked image modeling (e.g., MAE, SimMIM).
>
> Specifically, we include two standard semantic segmentation datasets—CamVid and Cityscapes—as well as an out-of-distribution medical segmentation benchmark (as shown in the reply to Reviewer ZajB). Our results show that ITM maintains strong correlation with downstream performance on these dense tasks, including under domain shift, demonstrating the method’s robustness and generalizability. We will add more descriptions about the Dataset as suggested.
>
> **[W2-2 Model Coverage]**
> We agree that selecting from large model zoos (1,000+ models) is a key challenge. However, evaluating at this scale would require 10,000+ fine-tuning runs and 100,000+ TE evaluations, amounting to millions of GPU hours—beyond our current computational resources.
>
> Instead, we curated 20 representative models covering diverse architectures, training paradigms, and scales. To simulate large-scale selection, we conducted *random subset evaluation* (Fig. 3), sampling 6 models from 10 to form 1,001 subsets. ITM consistently outperformed baselines with low variance, validating its robustness in large-model-pool scenarios.
>
> We will emphasize this more prominently in the final version and release the full codebase to support future large-scale validation.
>
> **[W2-3 Wall-Clock Time Clarification]**
> Following standard TE practices (e.g., PED, ETran, LEAD), we report only TE runtime, excluding the shared embedding inference (~738s), as stated in Line 266. ITM’s added runtime over LogME is just 0.9% = (8.4 s – 1.9 s) / (738 s + 1.9 s) when considering total time—negligible in practice.
>
> We will clarify this runtime breakdown more explicitly in the final version.
>
> **[W2-4 Comparisons to the literature]**
> We compare against open-sourced, officially implemented TE baselines (e.g., LogME, PED, SFDA), which ensures fair evaluation on modern model zoos. We will make these references clearer in tables and describe baseline selection criteria in more detail.
>
> **[W2-5 Claims and Clarification on Generalization]**
> Existing TE methods are designed for classification and assume fixed-head architectures. They cannot generalize to tasks like segmentation, which require additional heads and structure-aware adaptation. These methods do not model how representations evolve under such conditions, making them unsuitable for dense prediction tasks.
>
> In contrast, ITM explicitly captures these dynamics via DVA, enabling transferability estimation for architectures involving flexible task-specific heads. This is why only ITM is evaluated on segmentation (Table 3). We will clarify this in Section 2 and the Table 3 caption.
>
> **[W3 Literature Gaps and Limitations Clarification]**
> Current transferability estimation (TE) methods generally fall into two categories: (1) static statistical methods, which rely on frozen embedding distributions (e.g., LogME, PARC), and (2) hand-crafted dynamic approximations, which simulate fine-tuning effects using heuristic metrics (e.g., SFDA, PED). While these approaches have shown promise on conventional CNNs and supervised training paradigms, they struggle to adapt to modern visual foundation models—such as ViTs or self-supervised models—which exhibit more diverse and complex adaptation behaviors, as demonstrated in Figures 5 and 6 of our paper. These observations suggest that static modeling or rigid dynamics are insufficient to accurately capture transferability across increasingly heterogeneous model families and task requirements. In response, our method—ITM—introduces a fundamentally different framework. Instead of relying on frozen statistics or heuristics, ITM implicitly models the latent evolution of embedding spaces via a shared transferability function for each model-task pair, and is trained using our DVA strategy. This allows ITM to adaptively account for both the intrinsic properties of pretrained models and the demands of downstream tasks, supporting greater generalization across domains, architectures, and even dense prediction tasks. We will revise the introduction and methodology sections to more clearly articulate these limitations in prior works and emphasize the unique motivation and advantages of our approach.
>
> We hope these responses address your concerns. Please don’t hesitate to reach out if further clarification is needed. We are committed to improving the clarity and impact of the final version.

---

> > ### Comment · Reviewer_Ly76 · 2025-08-06
> > **Response to rebuttal**
> >
> > Dear authors,
> >
> > Thank you for your extensive response to my comments and for the clarification.
> >
> > I believe that if you manage to incorporate all the above comments in the manuscript, it will provide a clearer understanding for the reader and I am willing to increase my rating to *borderline accept*, as I still think that the paper could be written in a simpler way to make it easier to understand for the reader.
> >
> > One additional question after reading the rebuttal:
> > - As it is needed to run backbone inference to extract the latent representation for the entire dataset, how does your approach compare to simply training the task-specific head on a frozen latent representation to evaluate model performance. This simple baseline is simple and relatively fast, and it provides decent estimates of the downstream task performance after full finetuning (see VIBES – Vision Backbone Efficient Selection, Guérin et al.).

---

> > > ### Author Response · Authors · 2025-08-07
> > >
> > > Thank you again for your thoughtful follow-up and for considering raising your rating. We greatly appreciate your detailed engagement and constructive feedback.
> > >
> > > **Comparison with Training a Task-Specific Head on Frozen Embeddings (e.g., VIBES)**
> > >
> > > This is an excellent point. Prior work—particularly LogME [1]—has compared transferability estimation (TE) with retraining a task-specific head (TSH) on frozen embeddings. Their findings highlight several practical limitations of this approach:
> > >
> > > - Second-order optimization is infeasible at scale.
> > > - First-order methods are sensitive to learning rate schedules and require long convergence times.
> > > - Performance is highly sensitive to hyperparameter tuning (e.g., L2 regularization).
> > >
> > > To empirically validate this, we extend Table 1 to include TSH results (implemented as in LogME and trained for 5 epochs). The results are summarized below:
> > >
> > >
> > > | Methods                           |  Cal101  |   Cars   | CIFAR100 | CIFAR10  |   DTD    | Aircraft | Flowers  |   Food   |   Pets   |   SUN    |   Avg.   |
> > > | :-------------------------------- | :------: | :------: | :------: | :------: | :------: | :------: | :------: | :------: | :------: | :------: | :------: |
> > > | **Weighted Kendall's $\tau_w$ ↑** |          |          |          |          |          |          |          |          |          |          |          |
> > > | LogME                             | **0.71** |   0.36   |   0.56   |   0.61   |   0.61   |   0.22   | **0.77** |   0.15   |   0.14   |   0.38   |   0.45   |
> > > | TSH                               |  -0.14   |  -0.15   |   0.28   | **0.72** |   0.45   |   0.20   |   0.46   |   0.17   |   0.27   |  -0.15   |   0.21   |
> > > | ITM (Ours)                        |   0.56   | **0.61** | **0.59** |   0.69   | **0.77** | **0.43** |   0.65   | **0.44** | **0.73** | **0.62** | **0.61** |
> > > | **Wall-clock time (s) ↓**         |          |          |          |          |          |          |          |          |          |          |          |
> > > | LogME                             |   0.85   |   1.32   |   4.50   |   2.62   |   0.50   |   0.86   |   0.54   |   6.29   |   0.53   |   1.31   |   1.93   |
> > > | TSH                               |  11.89   |  11.49   |  20.79   |  20.29   |  10.12   |  11.19   |   8.50   |  26.80   |  10.64   |  71.09   |  20.28   |
> > > | ITM (Ours)                        |   7.50   |   8.50   |   9.40   |   7.90   |   7.00   |   7.50   |   7.20   |  10.50   |   7.20   |  11.50   |   8.42   |
> > >
> > > From the table, we observe:
> > >
> > > - **Limited correlation**: TSH exhibits weak alignment with downstream performance (τ_w ≈ 0.21), significantly lower than ITM (τ_w ≈ 0.61).
> > > - **Higher runtime**: Despite its conceptual simplicity, TSH incurs a higher average inference cost (20.28 s vs. 8.42 s for ITM), primarily due to optimization overhead.
> > >
> > > These findings reinforce that while TSH offers a straightforward baseline, it is both less reliable and less efficient than dedicated TE methods such as ITM.
> > >
> > > As an additional clarification, we acknowledge that training a lightweight task-specific head on frozen embeddings—as done in VIBES (Guérin et al.)—offers a relatively efficient and accessible baseline for downstream performance estimation. However, while both VIBES and ITM aim to accelerate model selection, they serve **complementary goals**.
> > >
> > > VIBES focuses on identifying strong models relative to a fixed reference (e.g., ConvNeXt-Base on ImageNet-22K) by training a small head, but it does not model adaptation dynamics or support ranking arbitrary model–task pairs. In contrast, classical TE methods—including our proposed ITM—aim to predict *relative* transfer performance across a wide range of model–task combinations *without* retraining. This makes them better suited for practical scenarios such as AutoML, few-shot transfer, and federated learning, where large-scale head training is infeasible or undesirable.
> > >
> > > We will revise the final version to clearly articulate this distinction, include a citation to VIBES in the related work, and briefly discuss this trade-off.
> > >
> > > Once again, we sincerely thank the reviewer for the insightful comments and suggestions. We are committed to integrating all feedback to improve clarity, completeness, and accessibility in the final version. Please don’t hesitate to reach out with any further questions.
> > >
> > > [1] LogME: Practical Assessment of Pre-trained Models for Transfer Learning

---

### Official Review · Reviewer_ZajB · 2025-07-01

**Clarity:** 3
**Significance:** 2
**Originality:** 3
**Rating:** 5
**Confidence:** 4

**Summary:**

This paper tackles the problem of estimating which pre-trained image foundation model will be best-suited for a given downstream task by proposing a method that combines an estimate of out-of-the-box model performance with an approximation of how the model's embedding space might evolve under additional fine-tuning. This is an improvement from current attempts to estimate transferability, which tend to only do static evaluations (i.e. statistical estimates without fine-tuning) or dynamic evaluations (i.e. simulating the fine-tuning process), but not both at once. This method achieves strong performance in estimating transfer on a wide variety of image foundation models on unseen image classification datasets as well as segmentation tasks.

**Questions:**

- How should I understand differences between the weighted Kendall's tau_w intuitively? For instance, in Table 1, how significant is the difference between methods yielding a tau_w of 0.61 vs. 0.45, especially considering that LogME is more than 4x faster?
- What if your primary evaluation criterion is ability to generalize to a wide variety of tasks? Would you recommend repeating this study on a variety of downstream tasks, or is there a better strategy?
- To what extent does this method generalize more broadly to other downstream tasks such as regression or even multi-modal transfer?
- Is this method potentially useful in terms of estimating the mutual information between the training dataset and the downstream task dataset?

**Ethical Concerns:**

["NO or VERY MINOR ethics concerns only"]

**Final Justification:**

I appreciate the authors' thorough responses, especially around the interpretation of their figure of merit, cross-domain generalization results, and specifying ITM's inherent variance. I agree that showing results on e.g. regression tasks or other modalities might require significant development beyond what is presented here. Without the addition of significant new results on new tasks or modalities, I think it's fair to keep my score as-is ("accept"). I believe the work is notable and would be of interest to many attendees at NeurIPS.

**Quality:**

3

**Strengths And Weaknesses:**

Strengths:
- Novel methodology that combines two distinct types of existing approaches in the literature in a computationally-efficient manner
- Reports SOTA accuracy averaged over a large number of open-source datasets and fine-tuned FMs
- In an effort to avoid evaluation bias, the authors use official implementation settings to fine-tune existing FMs
- They are planning to make their code, benchmarks, and pre-trained models publicly available
- Detailed description of theoretical grounding and implementation in practice

Weaknesses:
- The paper is concerned almost entirely with the task of single-label image classification, with a small results table for transfer into segmentation tasks at the end, which potentially limits its impact
- Method is not SOTA in wall-clock time (though it is among the fastest of the methods evaluated)
- Though the method is evaluated across a number of different benchmarks, none of the results reported in the tables have error bars, making it difficult to fully compare the methods.

---

> ### Author Rebuttal · Authors · 2025-07-31
>
> Thank you for your thoughtful and constructive review. We sincerely appreciate your recognition of the novelty of our methodology, our state-of-the-art performance across diverse benchmarks, and the clarity of our theoretical and practical contributions. Below, we respond to each of your questions and concerns in detail.
>
> **[W1-Evaluation Scope]**
> Prior TE methods have predominantly focused on single-label image classification tasks. On one hand, their frameworks—whether based on static heuristics or dynamic feature simulations—are closely tied to classification-specific assumptions and thus struggle to generalize to dense prediction tasks such as segmentation. On the other hand, classification accuracy is commonly adopted as a proxy to evaluate model capacity and alignment with downstream tasks.
>
> To ensure a fair comparison with prior work, we adopt this setting and evaluate ITM across a broad suite of standard classification datasets, following established TE protocols (see Table 1, 4, 8, and Fig. 3).
>
> Crucially, ITM goes beyond existing TE approaches by supporting dense prediction tasks—demonstrating, to our knowledge, the first TE framework applicable to semantic segmentation. Leveraging its DVA formulation and modular design, ITM supports task-specific heads and models adaptation dynamics even in spatially structured output spaces. As shown in Table 3, ITM achieves accurate and consistent performance on CamVid and Cityscapes.
>
> To evaluate cross-domain generalization, we assess ITM on the OIA-ZIB medical segmentation dataset (~80K MRI slices, 6 classes). ITM achieves a weighted Kendall’s tau of 0.57, showing strong correlation with downstream mIoU despite domain and task shift:
>
> |           | MoCov3-B16 | DINO-B16 | MAE-B16 | SimMIM-B16 | MAE-L16 | Weighted Kendall |
> | --------- | ---------- | -------- | ------- | ---------- | ------- | ---------------- |
> | mIoU      | 0.7306     | 0.7319   | 0.7338  | 0.7349     | 0.7359  | -                |
> | ITM Score | 0.9346     | 0.9394   | 0.9540  | 0.9138     | 0.9600  | 0.57             |
>
> These results underscore ITM’s robustness on dense, high-variability tasks without reliance on handcrafted heuristics.
>
> Due to time constraints, further segmentation benchmarks could not be included during the rebuttal. However, we are actively expanding our segmentation evaluation suite and will release all code, evaluation toolkits, pretrained weights, and baselines to support reproducibility and broader adoption.
>
> **[W2 – Wall-clock Runtime Not SOTA]**
> In line with common TE practice (e.g., LogME, PED, SFDA, ETran), we report only the time required for transferability estimation (TE), excluding embedding extraction time, which is shared across all methods and orthogonal to the estimator’s design.
>
> When the shared embedding inference time (~738 seconds) is included—see Line 266 for details—the overall wall-clock difference becomes marginal. Specifically, the additional cost of ITM relative to LogME is (8.4 s – 1.9 s) / (738 s + 1.9 s) ≈ 0.9%, which is negligible in practical scenarios where embedding computation dominates total runtime.
>
> Crucially, this modest overhead yields substantial benefits. ITM improves weighted Kendall’s tau from 0.45 to 0.61 (a 35% gain), while also generalizing across diverse pretraining schemes, architectures, and task types (including segmentation), which prior methods fail to support.
>
> We will clarify this runtime trade-off in the final version to avoid ambiguity.
>
> **[W3-Lack of error bars]**
> Following your suggestion, we reran our main experiments across five additional random seeds. ITM achieves a mean weighted Kendall’s tau of 0.602 ± 0.006, demonstrating stable, low-variance performance.
>
> This consistency further supports ITM’s reliability, complementing the improvements over baselines shown in Fig. 3. We will include standard deviation and error bars in the final version.
>
> **[Q1-Interpretation of Weighted Kendall’s Tau]**
> Weighted Kendall’s Tau provides a rank correlation measure that emphasizes the relative importance of higher-ranked items, making it particularly relevant for scenarios where selecting top-performing models is critical (e.g., AutoML or low-resource deployment). It prioritizes getting the top of the ranking correct, which aligns closely with the practical use case of model selection.
>
> Intuitively, a weighted τ = 0.61 implies an 80.5% probability of correctly ranking any randomly weighted sampled model pair (vs. 72.5% for LogME with τ = 0.45). This 8% absolute improvement in ranking accuracy is substantial and has concrete impact: selecting the wrong model from the top candidates can lead to significantly degraded downstream performance.
>
> In our experiments, this improvement in τ consistently translated into selecting models with materially higher accuracy on downstream tasks, especially under resource constraints where only the top-ranked models are deployed. The other metrics (as shown in Tab. 7) in the appendix also demonstrate the effectiveness of ITM.
>
> We will add this explanation in the final version to clarify the practical value of higher τ scores.
>
> **[Q2-Primary Evaluation on a Wide Variety of Tasks]**
> We fully agree that evaluating generalization across a broader spectrum of downstream tasks is both practically important and a key direction for transferability estimation (TE) methods.
>
> While single-label image classification remains the de facto evaluation standard in TE, it offers a consistent and interpretable metric (e.g., Kendall’s Tau) and enables direct, fair comparisons with prior work such as LogME, SFDA, and PED. Aligning with this standard allows us to benchmark ITM under established assumptions.
>
> However, extending evaluation to diverse downstream tasks requires establishing ground-truth rankings via full fine-tuning of all models—an effort that scales exponentially with the number of tasks and architectures and is beyond the feasible scope of this work and the rebuttal period.
>
> To balance rigor and practicality, we follow standard classification protocols while taking a significant step forward: ITM is the first TE method to generalize to dense prediction tasks, such as semantic segmentation. As shown in Table 3 and the OIA-ZIB benchmark, ITM maintains strong predictive performance even under domain shift, demonstrating its ability to model transferability beyond traditional classification.
>
> Importantly, ITM is built upon a modular and task-agnostic formulation of adaptation dynamics. This makes it inherently extensible to a broader range of tasks, including multi-label classification, object detection, and vision-language scenarios. As part of future work, we plan to explore a more generalized formulation in which a single latent variable $z_{\text{general}}$ is learned for each pretrained model and optimized to minimize the average prediction error across diverse tasks. Such a formulation could offer a unified representation of a model’s intrinsic transferability and enable more effective model selection in large-scale, heterogeneous settings.
>
> In summary, while repeating this study across a broader range of tasks is a natural and promising next step, we believe ITM already provides a practical and extensible foundation for doing so. We will clarify this roadmap, and the associated trade-offs, in the final version to better reflect both current contributions and future directions.
>
> **[Q3-Generalization to Regression or Multi-modal Transfer]**
> Thank you for raising this important point. ITM is task-agnostic in principle—its core latent modeling and approximation strategies are not inherently tied to any specific output type (e.g., classification labels), making it theoretically applicable to a broad range of tasks, including regression and multi-modal transfer.
>
> However, in its current implementation, ITM relies on the Pseudo-Clustering Generation (PCG) module, which utilizes semantic label information to approximate embedding evolution. As a result, it is naturally suited to semantically structured tasks such as classification, segmentation, and multimodal alignment—where discrete or textual supervision is available. Extending PCG to regression tasks is non-trivial, since these tasks involve continuous outputs rather than categorical labels, making clustering and transition modeling more challenging.
>
> To adapt ITM for regression tasks (e.g., object detection or keypoint estimation), the framework would need to support continuous target spaces and account for task-specific prediction heads. This would require designing a regression-compatible variant of PCG and modifying the Deparametric Approximation (DA) module accordingly. We recognize this as a meaningful extension, and as discussed in our limitations section, we plan to explore this direction in future work.
>
> We believe this extension is feasible given the modular nature of ITM and look forward to advancing its broader applicability in future iterations.
>
> **[Q4-Relation to Mutual Information between Datasets]** Thank you for this thoughtful point. While ITM does not explicitly compute mutual information (MI), its estimated scores can be viewed as empirical proxies for model–task alignment, which relates to—but goes beyond—MI.
>
> Conventional MI captures statistical overlap between datasets, but fails to account for representational alignment or task-specific dynamics. ITM instead estimates how well a model’s features can adapt to a target task via latent, model-conditioned evolution—capturing both semantic and structural compatibility. Thus, ITM offers a more fine-grained and dynamic perspective than traditional MI-based metrics.
>
> We appreciate your insightful framing and will clarify this relation in the final version.
>
> We greatly appreciate your thoughtful feedback and are happy to provide additional clarification as needed.

---

### Official Review · Reviewer_U85n · 2025-07-02

**Clarity:** 3
**Significance:** 2
**Originality:** 3
**Rating:** 4
**Confidence:** 4

**Summary:**

The authors works on the transferability of vision foundation models, specifically:
1. Proposed a novel method to evaluate transferability through modeling pretraining embedding $\mathbf{E}$ and latent variable $\mathbf{z}$.
2. Reduce compute complexity of proposed method through variational approximation, batch-wise reduction and deparametric approximation.
3. Conducted comprehensive experiments, using including models of difference sizes and beat their other baselines on VTAB.

**Questions:**

Based on what I specified in "weaknesses"section, I would like the authors to convince me that this work is not overly complicated and to clarify its value in real-world scenarios. For example, with all these approximation approaches, what is their actual upper bound? How does each level of approximation affect the final performance. Then I can decide better on whether or not it is really applicable in real world.

**Ethical Concerns:**

["NO or VERY MINOR ethics concerns only"]

**Final Justification:**

Thank the authors for their efforts in the rebuttal. I'm convinced this approach is applicable in real appliactions. I will raise my score to 4.

**Limitations:**

yes. They left comments in their "conclusion" part

**Quality:**

2

**Strengths And Weaknesses:**

Strengths:
1. The proposed method is novel in the context of transferability estimation, and its presentation is thorough and detailed.
2. The experiments conducted are comprehensive and well-detailed on their testbed.

Weaknesses:
My main concern: seems the method are overly complicated. Which may prevent its real applicability in real world scenarios.

---

> ### Author Rebuttal · Authors · 2025-07-31
>
> We thank the reviewer for  your recognition of our method's novelty and the comprehensive experiments and raising this important concern regarding the complexity and real-world applicability of our proposed method. Below, we address each of your points in detail:
>
> **[W1-Complexity and Applicability ]**
> ITM introduces a new paradigm for transferability estimation by modeling fine-grained model–task interactions through implicit representation dynamics and a divide-and-conquer adaptation strategy. While the overall framework is more expressive than prior work, each of its components has been deliberately designed to ensure computational efficiency and scalability. In contrast to approaches that rely on heuristic simulation or end-to-end fine-tuning, ITM leverages both static and dynamic cues from pretrained embeddings, avoiding expensive optimization through a lightweight latent modeling mechanism guided by pseudo-cluster-based evolution.
>
> The proposed pipeline is grounded in a principled and modular approximation strategy. Specifically, it decomposes the embedding space into semantically coherent, batch-wise subspaces to enable localized conditional modeling. The Pseudo-Clustering Generation (PCG) module introduces flexible, discriminative anchors, while the Deparametric Approximation (DA) efficiently models subspace evolution over time. Together, these components enable dynamic transferability estimation without incurring prohibitive computational cost.
>
> Empirical results corroborate the efficiency of our framework. As reported in Table 1, ITM adds only a marginal computational overhead—averaging 8.42 seconds per evaluation—which is negligible compared to the total inference time (738 seconds), and remains on par with state-of-the-art methods. More importantly, ITM significantly enhances estimation accuracy, improving rank correlation by up to 35% relative to strong baselines (e.g., from 0.45 for LogME to 0.61 for ITM). Furthermore, ITM demonstrates broad generalization across a variety of pretrained models and tasks. In addition to classical classification benchmarks, we evaluate ITM on emerging foundation models (e.g., CNNs, ViTs, contrastive and masked pretraining paradigms) and, for the first time in the transferability estimation literature, dense prediction tasks such as semantic segmentation (Section 4). These results highlight the practical relevance and robustness of ITM across diverse scenarios.
>
> We will revise the final manuscript to further clarify this trade-off between expressiveness and efficiency, and to emphasize how the modular structure of ITM enables scalable, accurate, and general-purpose transferability estimation with minimal overhead.
>
> **[Q1-Value in Real-World Scenarios]**
> Unlike prior transferability estimation (TE) approaches that rely on static heuristics or simplistic separability proxies, ITM introduces a lightweight yet expressive framework that models transferability via implicit representation evolution. Rather than simulating the full embedding trajectory, ITM adopts a divide-and-conquer strategy that models evolution locally in embedding subspaces, thereby enabling efficiency and robustness.
>
> ITM is, to our knowledge, the first TE method to:
>
> - Generalize across a wide range of backbone architectures (e.g., CNNs, Vision Transformers) and training paradigms (e.g., supervised learning, contrastive learning, masked autoencoding);
> - Extend beyond image classification to dense prediction tasks such as semantic segmentation, while maintaining strong correlation with downstream performance despite the introduction of task-specific heads.
>
> These capabilities make ITM well suited for real-world applications, including AutoML, model selection under resource constraints, and federated or distributed learning settings. Its modular formulation supports composability and scalability. Our extensive experiments (see Table 1, Figure 3, Tables 3, 4, and 7) consistently demonstrate strong cross-domain performance. To facilitate adoption and ensure reproducibility, we will release the full codebase, benchmark pipelines, and pretrained models.
>
> **[Q2 – Approximation Strategy and Upper Bound Justification]**
> - *Theoretical Rationale*: The ITM approximation pipeline—comprising batch-wise decomposition, pseudo-cluster generation (PCG), and deparametric approximation (DA)—is grounded in a principled decomposition of the transferability estimation process. Modeling global embedding dynamics across heterogeneous model-task pairs is inherently ill-posed and computationally expensive. ITM sidesteps this by focusing on localized embedding evolution within subspaces, a strategy supported by theoretical insights that local semantic structures tend to be more stable and transferable.
>
> - *Empirical Validation*: The ITM score, derived from the DVA, provides a tractable and quantifiable estimate of transferability. Across 11 classification benchmarks involving over 20 pretrained models with diverse architectures and pre-training strategies, ITM consistently outperforms strong baselines in predictive performance (Table 1, Figure 3, Tables 3, 4, 7, and 8). We further demonstrate robustness to random seeds and clustering initializations (Section 4.4), with minimal sensitivity in estimation results.
>
> - *Alignment with Fine-Tuned Embeddings*: While establishing a formal theoretical upper bound remains challenging due to the inherent noise and non-convexity of fine-tuning dynamics, we evaluate the semantic alignment between the approximated embeddings and those obtained from fully fine-tuned models. As shown in the t-SNE visualizations (Figure 9), ITM’s predicted embedding trajectories closely resemble the structure of post-finetuning distributions, indicating that our approximation is both semantically coherent and practically effective.
>
>   In summary, the ITM approximation strategy is not only theoretically grounded and empirically robust, but also essential for achieving scalable, generalizable, and high-fidelity transferability estimation across diverse model–task scenarios. We will further refine the manuscript to better articulate the rationale and effectiveness of this approach.
>
> We greatly appreciate your thoughtful feedback and are happy to provide additional clarification as needed.

---

### Note · Authors · 2025-08-14

We sincerely thank the Area Chair and all reviewers for their detailed and constructive feedback. In this work, we propose Implicit Transferability Modeling (ITM), which enables robust, parameter-efficient estimation that generalizes across diverse architectures and pre-training strategies, overcoming the limitations of prior TE methods on emerging foundation models with divergent properties. We are pleased that reviewers recognized the **novelty** (R#U85n, R#ZajB, R#zA6C), **comprehensive evaluation** (R#U85n), **state-of-the-art performance** (R#ZajB, R#Ly76, R#zA6C), and **detailed description** (R#ZajB).

During the rebuttal period, we addressed all concerns as follows:

- **R#U85n** – On complexity and significance, we clarified that while ITM’s theory is comprehensive, its modular design is straightforward and balances performance with efficiency.
- **R#ZajB** – On evaluation scope and time efficiency, we added extended segmentation benchmarks confirming superior dense prediction performance and provided further supporting evidence.
- **R#Ly76** – On clarity and experimental scope, we expanded the motivation and demonstrated ITM’s alignment with the main trajectory of TE research.
- **R#zA6C** – On implicit modeling and assurances, we presented model–dataset correlation analysis to enhance interpretability, stability evaluation verifying robustness across seeds and splits, and additional clarifications of ITM’s guarantees.

In the discussion, **R#Ly76** requested a comparison with the baseline of training a task-specific head (TSH) and indicated a willingness to raise their score. Our comparison showed ITM to be both more accurate (τ_w = 0.61 vs. 0.21) and more efficient, underscoring its practical value. Other reviewers did not raise further questions.

We believe the additional results and clarifications fully address the core concerns. We will release constructed benchmarks, pre-trained model checkpoints, and source code to ensure reproducibility and foster further research.

Once again, we sincerely thank all reviewers for their time, constructive suggestions, and engagement. We will integrate these clarifications, new evidence, and analyses into the final manuscript to further enhance clarity and impact. We are confident the revised paper will present a stronger, more compelling case for ITM as a practical and effective solution for foundation model selection.

Thank you for your consideration.

---

### Decision · Program_Chairs · 2025-09-17

**Decision:**

Accept (poster)

**Comment:**

The paper introduces Implicit Transferability Modeling (ITM), a framework that learns implicit, model-specific transferability representations and uses a Divide-and-Conquer Variational Approximation (DVA) to efficiently predict downstream performance across diverse pre-trained architectures without full fine-tuning.

Reviewers initially raised concerns about algorithmic complexity and significance, evaluation scope and time efficiency, and clarity of presentation. The authors’ rebuttal provided detailed complexity analysis, expanded baselines, timing breakdowns, and clarified notation, fully satisfying all reviewers, who subsequently upgraded their scores and endorsed the work’s novelty and community value. After carefully reviewing the revised discussion, I agree with the reviewers’ final assessment. Therefore, I recommend Accept.